# Mind Scramble: Unveiling Large Language Model Psychology Via Typoglycemia

## Abstract

Although still in its infancy, research into the external behaviors and internal mechanisms of large language models (LLMs) has shown significant promise in addressing complex tasks in the physical world. These studies suggest that powerful LLMs, such as GPT-4, are beginning to exhibit human-like cognitive abilities, including planning, reasoning, and reflection, among others. In this paper, we introduce an innovative research line and methodology named *LLM Psychology, which leverages or extends human psychology **experiments** and **theories** to investigate **cognitive** behaviors and mechanisms of LLMs.* Practically, we migrate the Typoglycemia phenomenon from psychology to explore the "mind" of LLMs. To comprehend scrambled text in Typoglycemia, human brains rely on context and word patterns, which reveals a fundamental difference from LLMs' encoding and decoding processes. Through various Typoglycemia experiments at the *character*, *word*, and *sentence* levels, we observe the following: **(I)** LLMs demonstrate human-like behaviors on a macro scale, such as slightly lower task accuracy with consuming more tokens and time; **(II)** Different LLMs show varying degrees of robustness to scrambled input, making it a democratized benchmark for model evaluation without crafting new datasets; **(III)** The impact of different task types varies, with complex logical tasks (*e.g.,* math) in scrambled format being more challenging. Going beyond these, some misleadingly optimistic results suggest that LLMs are **still primarily data-driven**, and their human-like cognitive abilities may differ from what we perceive; **(IV)** Interestingly, each LLM exhibit its *unique and consistent "cognitive pattern"* across various tasks, unveiling a general mechanism in its psychology process. To conclude, we provide an in-depth analysis of hidden layers on a micro scale to explain these phenomena, paving the way for LLMs' deeper interpretability and future research in *LLM Psychology*. [1]

## 1 Introduction

*"**Typoglycemia** refers to the pheonmneon where poeple can raed text even when the lettres in the midlde of wrods are scrambled, as long as the fisrt and last letters are in the crorect poistion."*

Do you notice that some words in the above explanation to Typoglycemia have letters in the wrong order? [pheonmneon, poeple, raed, ...] These words contain certain misplaced letters, yet we can still recognize them. This phenomenon, known as Typoglycemia, is widespread in human reading and is used in psychology experiments to study human language cognition (Johnson et al., 2007; Rayner et al., 2006). With recent development of large language models (LLMs), they demonstrate "human-like" capabilities and open a potential path for the upcoming artificial general intelligence, excelling in complex tasks such as tool using (Yuan et al., 2024), reasoning (Hao et al., 2023), planning (Kalyanpur et al., 2024), and role-playing (Chen et al., 2023a). However, research on the underlying cognitive mechanisms of LLMs remains in its infancy. Whether LLMs possess **deep** thinking and human-like cognition is an unsolved mystery that still looms over researchers (Binz & Schulz, 2023; Bender et al., 2021). Thus, we try to reveal this by exploring an intriguing question: *"Does LLM possess human-like cognitive processes and mechanisms in reading and comprehending?"*

To this end, this work aims to investigate the "human-like phenomena" demonstrated by LLMs and provide insights into whether these models truly possess cognitive capabilities or merely exhibit them in a statistical sense. To delve deeper into existing LLMs research, we categorize the off-the-shelf studies into three main research lines: **(I) Single-LLM**, where external human-like thought

---

[1]Our code is available at https://anonymous.4open.science/r/Typoglycemia-9DF1/

processes are applied to LLM through methods such as prompt engineering to achieve better performance (Liu et al., 2021). For instance, Wei et al. (2022); Yao et al. (2024); Besta et al. (2024) simulate human-like reasoning by guiding LLMs through intermediate thought steps in the structure of chain/tree/graph. **(II) Multi-LLMs (Agents)**, where interactions between multiple LLMs are used to explore their behavior and logic in complex communications, such as cooperative (Qian et al., 2023; Shen et al., 2024) and competitive (Zhao et al., 2023) scenarios, *etc.* **(III)** Notably, a small but growing body of work aims to investigate the intrinsic cognitive mechanisms of LLMs. Through cognitive science methods, Almeida et al. (2024) investigates LLMs' moral reasoning, while Zhang et al. (2023) study collaboration mechanisms among LLM-based agents.

However, **Line (I)** merely focuses on leveraging LLMs' human-like abilities to solve real-world problems, while overlooking deeper investigations into *why LLMs exhibit such capabilities*. This preconceived notion of equating LLMs with humans may overlook their limitations and misuse risks, leading to unreliable outcomes. **Line (II)**, constrained by specific parameters and settings, operates only within particular scenarios. This limitation results in reduced flexibility, as it prevents the agents from being adapted to diverse and unpredictable contexts beyond its predefined scope. Similarly, while **Line (III)** has begun to explore LLMs from a cognitive perspective, they remain limited to fixed and external scenarios such as moral reasoning

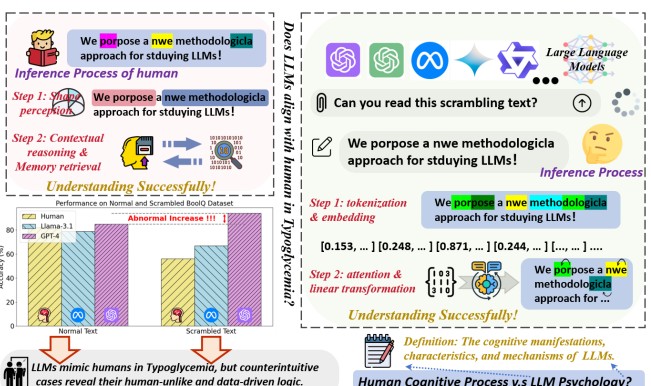

**Figure 1: (Upper Left)** The two-step process by which humans handle scrambled text. **(Lower Left)** The performance comparison among Human, Llama-3.1, GPT-4 on BoolQ dataset in both original and scrambled task description. We observe a widespread phenomenon of maintaining high accuracy, along with counter-intuitive improvements in certain cases. **(Right)** We draw the two-step process by which LLMs handle scrambled text to parallel with human. Scrambled text here is simple examples for better illustration. *See our practical scrambled text cases for experiments in Appendix G.*

or human simulation (Almeida et al., 2024; Petrov et al., 2024). What all of these studies lack is the systematical exploration of the generalized and intrinsic cognitive mechanisms of LLMs.

**Insights.** In this paper, we propose a new research line and concept: *LLM Psychology*, which follows and extends human psychology methods to explore and study LLMs. In practice, we use Typoglycemia as a lens to investigate the universal and underlying mechanisms of LLMs in comprehending. Psychologists, analyzing human behaviors in Typoglycemia scenarios, have explored human visual mechanisms, contextual reasoning, and language patterns (Agrawal et al., 2020; Caffarra et al., 2021). They discover that human reading relies on the overall shape of words and familiar patterns, enabling self-correction and holistic interpretation of scrambled text (Rayner et al., 2012). In a parallel vein, LLMs' tokenization algorithms, such as Llama's BPE (Sennrich, 2015; Touvron et al., 2023), shroud the inner mechanisms. Consequently, by applying Typoglycemia (**not transcription errors**) to LLMs, similar to what psychologists do with humans, we can explore whether LLMs demonstrate "human-like" performance and mechanisms from appearance to essence.

In practice, we first align humans with LLMs when processing scrambled text in Figure 1. We then naturally extend original Typoglycemia from *character* to *word* and *sentence* levels. To systematize subsequent study, we design the standardized experiment pipeline (referred as TypoPipe), which explores multi-dimensional performances through various tailor-made tasks in scrambled text(referred as TypoTasks). TypoPipe is deployed across 5 datasets on `Llama-3.1`, `Gemma-2` and `GPT` families. **Some interesting and counter-intuitive findings are as follows:** ♣ LLMs exhibit human-like behavior in TypoTasks, demonstrating a retained ability to comprehend scrambled text, albeit at a higher computational cost. ♦ The emergent human-like abilities of LLMs are fundamentally statistical and data-driven, rather than genuinely resembling human cognition. As shown in the lower-left portion of Figure 1, `GPT-4` displays an abnormal improvement ($0.2 \sim 2.2\%$ ↑) on scrambled text that is typically more challenging. ♥ Further experiments reveal a strong correlation between hidden layer semantics and model performance, indicating that transformers' focus on certain displaced information in scrambled text may drive this unexpected improvement statistically. ♠ Each LLM ex-

hibits its unique and consistent hidden layer semantics distribution across different Typoglycemia tasks. This mirrors how individual humans possess their own unique cognitive patterns.

In summary, our core contributions can be listed as follows:

❶ *New Direction.* We propose "*LLM Psychology*" as an interdisciplinary framework with significant research depth, offering novel methodologies, directions and insights for the future study of LLM's human-like cognition. To the best of our knowledge, we are the pioneer to systematically transfer cognitive psychology methodologies and experiments to LLMs, assessing the similarities and differences between LLMs and humans from a cognitive psychological perspective.

❷ *Comprehensive Experiments.* We extend the original Typoglycemia experiments in psychology and adapt them to LLMs, using tailor-designed TypoPipe and TypoTask frameworks, we conduct extensive experiments on 8 models across 5 datasets, testing over 20 types of scrambled text at *character*, *word*, and *sentence* levels with distinct reordering, inserting, and deleting operations. Our results align LLMs' performance and cost changes with human behaviors in these scenarios.

❸ *Deep Analysis.* We report LLMs' *unique "cognitive pattern"* and *anomalous behaviors*. We explore the underlying causes through an analysis of hidden layer semantics in the encoder and decoder. Our findings demonstrate that LLMs' emergent human-like abilities are driven by data and statistics, providing strong evidence that their "cognitive process" differ from that of humans.

❹ *Democratized Benchmark.* We present an innovative, implementable, yet effective benchmark based on the Typoglycemia method to evaluate LLMs' capabilities *based on existing datasets*. Our experimental results reveal varying degrees of robustness across different LLMs, validating that our benchmark correlates well with commonly accepted assessments of their ability.

## 2 RELATED WORK

**Human-like Mechanisms of LLMs.** LLMs (Touvron et al., 2023; Achiam et al., 2023; Wang et al., 2024) have revolutionized both academic and industrial research paradigms, owing to their exceptional and human-like capabilities (Wei et al., 2022; Wang et al., 2024). Building on these foundational studies, numerous research efforts integrate mechanisms such as memory, role-playing and tool using to fully leverage these human-like capabilities (Wei et al., 2022; Bubeck et al., 2023; Hong et al., 2023; Li et al., 2023; Chen et al., 2023b;a). Several studies explore the similarities between LLMs and human cognitive mechanisms. McCoy et al. (2019) investigates model's cognitive intuitions in reasoning tasks. Chowdhery et al. (2023) analyzes PaLM's memory mechanisms, highlighting its human-like processes in retrieval and question answering. Though promising, there is still a debate that whether LLMs truly understand language or merely rely on data-driven patterns (Bender et al., 2021). *We use Typoglycemia as a psychological probe to uncover the superficial performance and underlying mechanisms of LLMs and present "LLM Psychology" with the first shot.*

**Cognitive Concepts in LLMs.** Due to the similarity between LLMs and humans, many studies have been inspired by human cognitive process to enhance LLMs (Bubeck et al., 2023; Wei et al., 2022). For example, SwiftSage (Lin et al., 2024) improves the task capabilities of LLM-based agents in dynamic environments by mimicking the dual-process theory of human cognition. Xie et al. (2024) investigates whether LLMs can simulate human cognitive trust behaviors by employing a series of trust games. AvalonBench (Light et al., 2023) evaluates LLMs' competency levels through the Resistance Avalon game, which involves cognitive strategies. PsySafe (Zhang et al., 2024) further explores the impact of cognitive states as prompts on the safety of LLM-agent. In this work, inspired by the *Typoglycemia psychological experiment*, we explore the underlying cognitive process of LLMs by comparing their micro and macro level performance with that of humans.

**Human Reading and Typoglycemia.** An interesting phenomenon is that humans can maintain an understanding of the general meaning of scrambled text, a capability that some studies attribute to the brain's mechanisms of holistic shape perception and pattern recognition (Miller, 1994; Rayner et al., 2006; Perea & Lupker, 2004; Shaywitz & Shaywitz, 2008; Rayner et al., 2012). As LLMs' powerful understanding capabilities have been recognized, a few studies attempt to explore whether LLMs exhibit similar "human-like" phenomena. Cao et al. (2023) investigates the exceptional performance of LLMs in reconstructing character-level scrambled text. Singh et al. (2024) finds that LLMs can still maintain encoding consistency when confronted with such text. However, previous work has merely showcased related phenomena without delving deeply into the underlying mechanisms of LLMs. In our research, we systematically migrate the Typoglycemia phenomenon across multi-granularity to LLMs and provide a comprehensive explanation for its underlying causes.

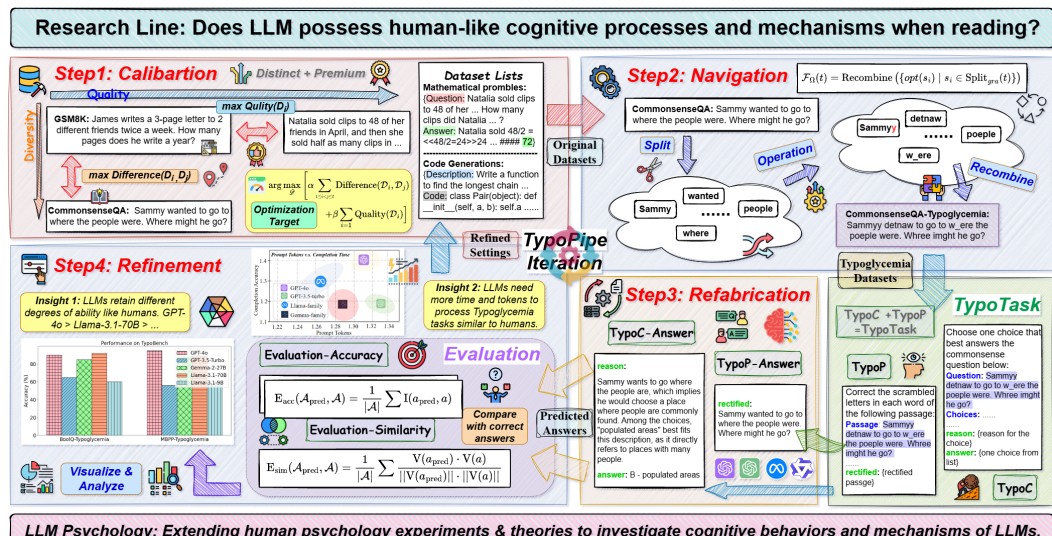

**Figure 2:** *TypoBench Overview*. TypoPipe and TypoTask form the two components of our benchmark. The overall pipeline consists of 4 steps: **Calibration**, **Navigation**, **Refabrication**, and **Refinement**. TypoTask consists of two task categories: **TypoC** and **TypoP** which emphasize performance and perception, respectively.

## 3 CAN LLMS RECOGNIZE TPYOGYLCMEIA AS TYPOGLYCEMIA?

To apply core principles of *LLM Psychology*, we migrate and extend the Typoglycemia concept from psychology by proposing the calibrated benchmark (**TypoBench**), as shown in Figure 2 . Concretely, TypoBench consists of two components: (1) Typoglycemia Pipeline that provides standardized experiment workflow on LLMs (Sec 3.1) and (2) Typoglycemia Task that challenges LLMs' all-around abilities to address scrambled text (Sec 3.2), with reasons for its specific design in Sec 3.3.

### 3.1 TYPOGLYCEMIA PIPELINE (TYPOPIPE)

In this section, we introduce the generalized framework TypoPipe to standardize the experimental process. TypoPipe divides the entire pipeline into the following 4 steps: ① **Calibration** aims to comprehensively select and calibrate datasets for a thorough evaluation of LLMs' ability. ② **Navigation** targets to design reasonable functions (TypoFunc) to transform each data into various types of "Typoglycemia" text. ③ **Refabrication.** This process perform the original task on the dataset or design other scenarios to explore LLMs' versatile performances. ④ **Refinement** consists of iteratively calculating metrics, analysing results and refining experiment settings for final conclusions.

**Formulations.** First, we provide denotations for further formulations. Let $\mathcal{C}$ be the character set, then the text set is $\mathcal{T} = \{c_1 c_2 \dots c_n \mid c_i \in \mathcal{C}, 1 \le i \le n\}$. Then we denote a dataset with questions and answers as $\mathcal{D} = (\mathcal{Q}, \mathcal{A}) = \{(q_i, a_i) \mid q_i \in \mathcal{Q}, a_i \in \mathcal{A}, 1 \le i \le m\}$, and LLM as a function $M : \mathcal{T} \to \mathcal{T}$. For any finite set $\mathcal{X}$, we use $x_i (1 \le i \le |\mathcal{X}|)$ to refer to its element for convenience.

① *Calibration*: We denote the family of datasets as $\mathscr{D} = \{\mathcal{D}_1, \mathcal{D}_2, \dots, \mathcal{D}_t\}$. Calibration aims at:

$$\arg\max_{\mathscr{D}} \left[ \alpha \sum_{1 \le i < j \le t} \text{Difference}(\mathcal{D}_i, \mathcal{D}_j) + \beta \sum_{i=1} \text{Quality}(\mathcal{D}_i) \right] \quad (1)$$

Eq 1 seeks to select distinct and premium datasets to challenge LLMs from multi-aspects. In practice, we heuristically select tailored datasets from distinct fields. (See in Appendix B.1).

② *Navigation*: Indicate the binary set $(opt, gra)$ as $\Omega$, where $opt : \mathcal{T} \to \mathcal{T}$ is the text operation (reorder, insert, delete, etc.) and $gra \in \{\text{character}, \text{word}, \text{sentence}\}$ is the smallest operational unit (granularity) for $opt$. We define TypoFunc $\mathcal{F}_\Omega : \mathcal{T} \to \mathcal{T}$, where $\forall t \in \mathcal{T}$,

$$\mathcal{F}_\Omega(t) = \text{Recombine}\left(\{opt(s_i) \mid s_i \in \text{Split}_{gra}(t)\}\right), \quad (2)$$

where function $\text{Split}_{gra} : \mathcal{T} \to \mathcal{T}^*$ maps text into a set of tokens split at the specified granularity level and function $\text{Recombine} : T^* \to T$ recombines tokens into text.

③ *Refarbrication*: Based on datasets and functions from previous two steps, we then apply them to get the Typoglycemia prompts. Concretely, let $\mathrm{P} : \mathcal{T} \to \mathcal{T}$ be the function that transforms data into prompts under certain task scenarios (See examples in Appendix C). For any function $f$ and set $\mathcal{X} = \{x_1, x_2, \ldots, x_{|\mathcal{X}|}\}$, denote $f(\mathcal{X}) = \{f(x_1), \ldots, f(x_{|\mathcal{X}|})\}$ as applying $f$ to all individual elements in $\mathcal{X}$ respectively. For any dataset $\mathcal{D} = (\mathcal{Q}, \mathcal{A})$, we define refarbrication step $\mathcal{T} \to \mathcal{T}$ as:

$$\mathcal{P} = \mathrm{P}\big(\mathcal{F}_\Omega(\mathcal{Q})\big) \;\mapsto\; \mathcal{A}_{\mathrm{pred}} = \mathrm{M}(\mathcal{P}) \tag{3}$$

Eq 3 expresses the process of converting the original texts in dataset to the Typoglycemia text tasks and get corresponding responses from LLMs. $\mathcal{A}_{\mathrm{pred}}$ is the LLMs' solutions or answers to inputs.

④ *Refinement*: After step ③, we utilize evaluation function $\mathrm{E}^{\mathcal{D} \to \mathbb{R}}$ to quantify LLMs' performance on corresponding tasks. The accuracy evaluation metrics is:

$$\mathrm{E}_{\mathrm{acc}}(\mathcal{A}_{\mathrm{pred}}, \mathcal{A}) = \frac{1}{|\mathcal{A}|} \sum \mathrm{I}(a_{\mathrm{pred}}, a), \quad \text{where} \quad \mathrm{I}(x, y) = \begin{cases} 1, & \text{if } x = y \\ 0, & \text{otherwise} \end{cases} \tag{4}$$

Here $\mathrm{E}_{\mathrm{acc}}$ represents the accuracy between LLMs' answers and correct answers. Since accuracy only evaluates the final results instead of **intermediate** thinking process of LLMs, we import a new metric to asses the semantic similarity of hidden states and representations in Transformers. Denote the embedding function as $\mathrm{V} : \mathcal{T} \to \mathbb{R}^d$. The semantic similarity evaluation metrics is:

$$\mathrm{E}_{\mathrm{sim}}(\mathcal{A}_{\mathrm{pred}}, \mathcal{A}) = \frac{1}{|\mathcal{A}|} \sum \frac{\mathrm{V}(a_{\mathrm{pred}}) \cdot \mathrm{V}(a)}{||\mathrm{V}(a_{\mathrm{pred}})|| \cdot ||\mathrm{V}(a)||}, \tag{5}$$

where $x \cdot y$ denotes the dot product of vectors and $||z||$ denotes the Euclidean norm of vector. $\mathrm{E}_{\mathrm{sim}}$ assesses the cosine similarity of LLMs' output with standard answers from a semantic view.

Finally, for any dataset $\mathcal{D} = (\mathcal{Q}, \mathcal{A})$, a complete iteration of TypoPipe is represented as:

$$\mathrm{TP}(\mathcal{D}, \mathcal{F}_\Omega, \mathrm{M}, \mathrm{P}, \mathrm{E}) = \mathrm{E}\Big[\mathrm{M}\big(\mathcal{F}_\Omega(\mathrm{P}(\mathcal{Q}))\big), \mathcal{A}\Big] \tag{6}$$

TP is the function representing the whole TypoPipe. Going beyond this, we propose some metrics to evaluate LLMs' ability from the Typoglycemia perspective:

$$\mathbb{T}_{\mathrm{abs}} = \sum_{i=1}^{u} \alpha_i \cdot \mathrm{TP}(\mathcal{D}_i, \mathcal{F}_\Omega, \mathrm{M}, \mathrm{P}, \mathrm{E}) \text{ or } \sum_{j=1}^{v} \alpha_j \cdot \mathrm{TP}(\mathcal{D}, \mathcal{F}_{\Omega_j}, \mathrm{M}, \mathrm{P}, \mathrm{E}) \tag{7}$$

$$\mathbb{T}_{\mathrm{rel}} = \sum_{i=1}^{w} \alpha_i \cdot \frac{\mathrm{TP}(\mathcal{D}_i, \mathcal{F}_\Omega, \mathrm{M}, \mathrm{P}, \mathrm{E})}{\mathrm{TP}(\mathcal{D}_i, \mathcal{F}_\ddagger, \mathrm{M}, \mathrm{P}, \mathrm{E})}, \tag{8}$$

where $\sum \alpha = 1, \alpha \in [0, 1]$ and $\mathcal{F}_\ddagger$ refers to the identity transformation. $\mathbb{T}_{\mathrm{abs}}$ and $\mathbb{T}_{\mathrm{rel}}$ evaluate LLM's absolute and relative performances on various datasets or TypoFuncs, respectively.

## 3.2 Typoglycemia Task = TypoC + TypoP

Building upon the standardized TypoPipe workflow, we have carefully designed TypoTask, which targets at assessing LLMs' performance in specific Typoglycemia-related tasks, along with their ability to comprehend and correct scrambled text. Specifically, TypoTask consists of the following two categories of tasks: Typoglycemia Completion (**TypoC**) and Typoglycemia Perception (**TypoP**).

**TypoC** refers to performing native tasks on the dataset. For example, the native task of GSM8K (Hendrycks et al., 2020) is to solve mathematical problems. TypoC reflects LLMs' ability to comprehend and follow scrambled text prompt when addressing problems in specific fields. To further explore the extent to which LLMs understand Typoglycemia text (scrambled text), we design **TypoP** consisting of *Rectify*, *Summarize*, and *Translate*. *Rectify* task aims to restoring Typoglycemia text back to its original form, assessing the model's ability to locally identify and rectify such errors. *Summarize* and *Translate* tasks require summarizing and translating, respectively, which evaluates the model's ability to understand the global context and detailed information in Typoglycemia text. See tailor-selected TypoC and TypoP tasks in Appendix B, C.1, and C.2.

### 3.3 WHY COMPLETION AND PERCEPTION?

Methodologically, TypoC is designed to evaluate the behavioral performance of LLMs, while TypoP aims to assess their perception and understanding, drawing inspiration from behavioral psychology and cognitive psychology, respectively. In doing so, we provide a vivid example of how psychological principles can be applied to understand and evaluate LLMs via our proposed *"LLM Psychology"*. These two tasks explore the impact of Typoglycemia on LLMs from both **fine-grained** and **coarse-grained** perspectives, progressing from shallow to deep levels of analysis. To successfully complete these tasks, models must simultaneously grasp local (scrambled content) and global information (contextual semantics) in order to fully comprehend the task's details and objectives. Based on the TypoBench framework, Eq 7 and Eq 8, we propose a more general and concise method for evaluating LLMs based on existing datasets, which reflects abilities not explored in previous research:

$$\mathbb{T}_{\text{gen}} = \frac{\text{E}(\text{M}, \mathcal{F}(\mathcal{D}))}{\text{E}(\text{M}, \mathcal{D})} \tag{9}$$

Eq 9 means using metrics E to evaluate model M on dataset $\mathcal{D}$ before and after being applying function $\mathcal{F}$. We present division to quantify the impact of $\mathcal{F}$. In our work, $\mathcal{F}$ is Typoglycemia.

## 4 EXPERIMENT

We employ TypoPipe across various scenarios to comprehensively study the impact of Typoglycemia on LLMs. The experiments are designed to investigate the following research questions:

- RQ1: What is the impact of Typoglycemia on existing LLMs?
- RQ2: How do other Typoglycemia Functions (*e.g.,* insertion and deletion) impact LLMs?
- RQ3: What are the effects of increasing the scrambling ratio of Typoglycemia?
- RQ4: Why do LLMs align with human performance under Typoglycemia?

### 4.1 EXPERIMENTAL SETUPS

**Datasets.** We aim to evaluate LLM Psychology across various task settings, including *mathematics, code generation, situational question answering*, and *commonsense reasoning*. Concretely, as for scenarios requiring strong logical reasoning, we select GSM8k (Hendrycks et al., 2020) for math and MBPP (Kocetkov et al., 2022) for code. Additionally, we explore the impact of Typoglycemia on LLMs' emergent situational learning and knowledge capabilities. We select BoolQ (Clark et al., 2019) and SQuAD (Rajpurkar et al., 2016) dataset for situational question answering tasks. For commonsense reasoning, we use CSQA (Talmor et al., 2018) dataset, a multiple-choice commonsense dataset. More descriptions on dataset can be found in Figure 1, Appendix B, and C.

**TypoFuncs** ($\mathcal{F}_\Omega$) transform the above datasets into Typoglycemia texts. To extend psychological Typoglycemia, we execute $\mathcal{F}_\Omega$ at character, word, and sentence levels, allowing us to explore the sensitivity of LLMs to various text variations. Specific $\mathcal{F}_\Omega$ operations include reordering, inserting, and deleting (refer to as **REO**, **INS**, and **DEL**, respectively). Operation X can be applied in different positions or ways of the three levels, such as: all (**X-ALL**), internal (**X-INT**), adjacent (**X-ADJ**), beginning (**X-BEG**), ending (**X-END**), and reversing (**X-REV**). Utilizing well-designed $\mathcal{F}_\Omega$, our Typoglycemia experiment contains both mildly scrambled text and highly disordered text that is nearly unrecognizable to humans. The specific operations instances can be found in Appendix D.

**Models and Metrics.** We extensively evaluate our concept across diverse LLMs within **zero-shot** setting, including **Gemma-2** (2B, 9B and 27B) (Team et al., 2024), **Llama 3.1** (8B, 70B) (Touvron et al., 2023), **GPT-3.5-Turbo**[2], **GPT-4o-mini**[3] and **GPT-4o**[4]. The selection of these models and their corresponding sizes provides a comprehensive "model zoom". In our settings, we choose accuracy and cosine similarity as metrics. For accuracy, we consider a response correct *only* when the LLM's output *exactly* matches the correct answer. For cosine similarity, we embed the reasoning processes into vectors using the **text-embedding-3**[5] and calculate cosine similarity with the standard process. The model parameter settings for reproducibility can be found in Appendix E.

[2]https://platform.openai.com/docs/models/gpt-3-5-turbo
[3]https://platform.openai.com/docs/models/gpt-4o-mini
[4]https://platform.openai.com/docs/models/gpt-4o
[5]https://platform.openai.com/docs/models/embeddings

**Table 1: Main results on the TypoC tasks when $\mathcal{F}_\Omega$ = REO on the *character*, *word* and *sentence* level.** We evaluate the average task accuracy (over 3 runs) of various LLMs on the GSM8k, BoolQ, and CSQA datasets. **BASE** refers to the scenario where $\mathcal{F}_\Omega$ is not applied to the task description. With the same coefficient weights, $\mathbb{T}_{\text{abs}}$ (Eq 7) shows each row's average accuracy, evaluating general performance across various TypoFuncs. $\mathbb{T}_{\text{rel}}$ calculates Eq 8, quantifying the ability retaining ratio compared with BASE. In each dataset, red (blue) marks the maximum value in each row (column), and green marks values that are the maximum in both. Gray marks the values that are higher than BASE in each row. Several TypoC cases are shown in Appendix G.

| Datasets/Models | Standard | Character | | | | | Word | | | Sentence | | | $\mathbb{T}_{\text{abs}}/\mathbb{T}_{\text{rel}}$ |
|---|---|---|---|---|---|---|---|---|---|---|---|---|---|
| | BASE | ALL | INT | BEG | END | REV | ALL | ADJ | REV | ALL | ADJ | REV | |
| **GSM8k**: *A dataset of grade-school-level mathematical problems with multi-step solutions.* | | | | | | | | | | | | | |
| Gemma-2-2B | 59.3 | 6.5 | 29.8 | 31.0 | 40.3 | 1.3 | 7.3 | 30.5 | 7.2 | 38.5 | 47.8 | 31.8 | 24.7/41.7% |
| Gemma-2-9B | 86.5 | 30.8 | 73.8 | 78.5 | 84.3 | 2.3 | 37.0 | 68.5 | 46.5 | 77.3 | 79.0 | 70.3 | 58.9/68.1% |
| Gemma-2-27B | 87.8 | 36.0 | 78.3 | 82.3 | 86.0 | 3.1 | 39.3 | 73.3 | 48.8 | 77.4 | 81.0 | 74.0 | 61.7/70.2% |
| Llama-3.1-8B | 84.5 | 15.8 | 59.5 | 62.8 | 76.0 | 1.5 | 25.0 | 64.3 | 30.8 | 69.0 | 77.8 | 65.5 | 49.8/58.9% |
| Llama-3.1-70B | 97.0 | 51.3 | 88.0 | 93.5 | 94.5 | 3.6 | 54.3 | 82.5 | 64.8 | 89.3 | 89.3 | 84.8 | 72.4/74.6% |
| GPT-3.5-Turbo | 77.0 | 32.8 | 64.0 | 68.0 | 71.8 | 5.8 | 34.1 | 60.3 | 40.1 | 67.3 | 70.1 | 65.0 | 52.6/68.3% |
| GPT-4o-mini | 90.3 | 45.0 | 79.2 | 82.0 | 86.0 | 25.7 | 41.0 | 75.5 | 51.2 | 78.5 | 81.8 | 78.0 | 65.8/72.9% |
| GPT-4o | 91.8 | 82.7 | 89.8 | 89.3 | 91.5 | 68.3 | 56.3 | 85.0 | 72.5 | 82.8 | 85.0 | 83.2 | 80.6/87.8% |
| **BoolQ**: *A question-answering dataset consists of yes/no questions and corresponding context passages.* | | | | | | | | | | | | | |
| Gemma-2-2B | 75.7 | 68.7 | 70.7 | 75.2 | 74.0 | 52.7 | 72.2 | 74.5 | 72.8 | 75.3 | 75.7 | 74.7 | 71.5/94.5% |
| Gemma-2-9B | 88.5 | 79.0 | 85.7 | 88.5 | 88.2 | 69.3 | 83.8 | 87.2 | 84.5 | 86.8 | 88.3 | 89.0 | 84.6/95.6% |
| Gemma-2-27B | 89.3 | 86.8 | 88.3 | 85.5 | 86.8 | 69.0 | 84.8 | 89.5 | 84.7 | 90.5 | 90.5 | 87.3 | 85.8/96.1% |
| Llama-3.1-8B | 84.2 | 71.3 | 79.3 | 83.3 | 82.8 | 66.0 | 79.0 | 80.7 | 77.2 | 83.3 | 84.2 | 83.5 | 79.1/93.9% |
| Llama-3.1-70B | 90.5 | 81.5 | 88.8 | 89.0 | 90.2 | 69.7 | 85.7 | 89.8 | 86.5 | 90.7 | 90.3 | 89.3 | 86.5/95.6% |
| GPT-3.5-Turbo | 86.2 | 71.8 | 78.8 | 82.7 | 83.8 | 65.0 | 71.3 | 79.5 | 73.2 | 82.5 | 82.7 | 82.2 | 77.6/90.0% |
| GPT-4o-mini | 88.7 | 83.2 | 87.0 | 89.3 | 90.3 | 80.7 | 85.2 | 89.2 | 86.7 | 89.7 | 89.5 | 88.3 | 87.2/98.3% |
| GPT-4o | 91.3 | 91.3 | 91.8 | 92.5 | 93.5 | 92.7 | 92.0 | 93.1 | 91.8 | 92.2 | 92.0 | 92.2 | 92.3/101.1% |
| **CSQA**: *A multiple-choice question dataset based on everyday knowledge.* | | | | | | | | | | | | | |
| Gemma-2-2B | 57.9 | 25.2 | 42.1 | 45.1 | 50.2 | 20.9 | 45.0 | 51.7 | 39.7 | 59.5 | 59.0 | 58.9 | 45.2/78.1% |
| Gemma-2-9B | 69.7 | 34.8 | 56.7 | 61.5 | 65.7 | 23.5 | 54.8 | 61.6 | 50.9 | 69.0 | 69.9 | 68.8 | 56.1/80.5% |
| Gemma-2-27B | 70.7 | 37.5 | 57.3 | 60.6 | 66.7 | 29.3 | 54.2 | 63.9 | 52.5 | 70.0 | 69.7 | 69.6 | 57.4/81.2% |
| Llama-3.1-8B | 66.6 | 29.4 | 43.6 | 48.8 | 58.0 | 20.9 | 49.4 | 57.1 | 47.3 | 67.1 | 66.5 | 66.4 | 50.4/75.7% |
| Llama-3.1-70B | 73.7 | 39.8 | 60.4 | 65.4 | 67.9 | 28.7 | 57.8 | 67.3 | 57.9 | 73.4 | 73.4 | 73.1 | 60.5/82.1% |
| GPT-3.5-Turbo | 67.0 | 37.4 | 57.4 | 58.9 | 62.0 | 28.5 | 51.7 | 59.2 | 48.0 | 67.7 | 66.0 | 66.9 | 54.9/81.9% |
| GPT-4o-mini | 73.5 | 40.0 | 58.8 | 63.5 | 66.7 | 47.4 | 53.9 | 64.8 | 53.7 | 72.8 | 72.2 | 72.5 | 60.6/82.4% |
| GPT-4o | 75.7 | 56.7 | 70.3 | 73.3 | 73.9 | 65.1 | 62.8 | 70.8 | 63.6 | 75.0 | 76.6 | 76.1 | 69.5/91.8% |

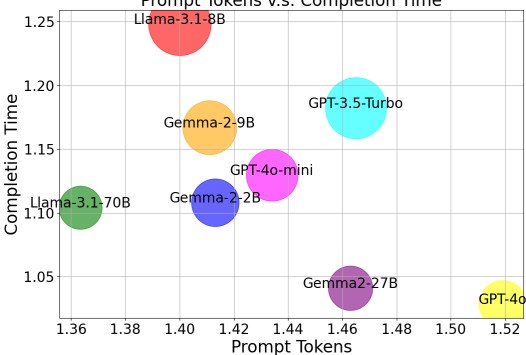

**Figure 3: Token and time consumption ratio** before and after being processed by TypoFunc when $\mathcal{F}_\Omega$ = REO-INT on *character* level for BoolQ dataset.

| Models | ALL | INT | BEG | END | REV | |
|---|---|---|---|---|---|---|
| Gemma-2-2B | 16.6 | 53.2 | 71.2 | 87.1 | 5.0 | ✓ |
| Gemma-2-9B | 30.1 | 86.8 | 87.1 | 91.4 | 11.7 | ✓ |
| Gemma-2-27B | 47.8 | 91.4 | 93.9 | 96.1 | 25.1 | ✓ |
| Llama-3.1-8B | 14.6 | 54.7 | 74.7 | 87.1 | 4.8 | ✓ |
| Llama-3.1-70B | 39.5 | 85.9 | 92.9 | 95.2 | 16.4 | ✓ |
| GPT-3.5-Turbo | 72.8 | 95.4 | 96.6 | 96.9 | 68.1 | ✓ |
| GPT-4o-mini | 68.3 | 94.5 | 96.8 | 95.8 | 80.7 | |
| GPT-4o | 93.8 | 97.5 | 97.3 | 97.8 | 95.3 | ✓ |

**Table 2: Results (Accuracy) on TypoP-Rectify task** when $\mathcal{F}_\Omega$ = REO on *character* level for GSM8k. ✓ means the accuracy ranking is similar to that of TypoC. See Rectify cases in Appendix H.1.

## 4.2 MAIN RESULTS (RQ1)

To answer RQ1, we compare different Typoglycemia concepts across various models and datasets. We apply ***random*** reordering and run the experiments multiple times, reporting the mean values. The experimental observations (**Obs**) are as follows and experiment discussion is placed in Appendix A:

**Obs.1. Typoglycemia generally leads to a decline in model performance, with more advanced models being less affected.** As shown in Table 1, red markers predominantly appear in the BASE column, indicating that accuracy tends to decrease after applying $\mathcal{F}_\Omega$. The performance retention of models within the same series increases with model size. For instance, on GSM8k dataset, the Gemma-2 series exhibits an increase in average accuracy across scales, with retention rates

**Table 3: Results on the TypoC tasks when $\mathcal{F}_\Omega =$ INS and DEL on *character* levels.** We apply $\mathcal{F}_\Omega$ at the begin and end of each word. We report the average accuracy (over 3 runs) of various LLMs on the GSM8k, BoolQ, and CSQA datasets. **BASE** means $\mathcal{F}_\Omega$ is not applied. $\mathbb{T}_{\text{abs}}$ shows each column's average accuracy, while $\mathbb{T}_{\text{rel}}$ calculates our proposed metrics with equal weights. In each dataset, red marks the maximum in columns. Gray marks the values that are higher than BASE in columns. TypoC cases are in Appendix G.

| Datasets/$\mathcal{F}_\Omega$ | Gemma-2-2B | Gemma-2-9B | Gemma-2-27B | Llama-3.1-8B | Llama-3.1-70B | GPT-3.5-Turbo | GPT-4o-mini | GPT-4o |
|---|---|---|---|---|---|---|---|---|
| **GSM8k** | | | | | | | | |
| BASE | 59.3 | 86.5 | 87.8 | 84.5 | 97.0 | 77.0 | 90.3 | 91.8 |
| INS-BEG | 42.0 | 76.3 | 87.0 | 75.5 | 94.8 | 70.8 | 87.5 | 90.3 |
| INS-END | 42.8 | 87.0 | 84.0 | 74.3 | 95.0 | 70.0 | 87.0 | 90.8 |
| DEL-BEG | 37.3 | 79.5 | 83.0 | 63.8 | 91.8 | 69.0 | 82.3 | 90.8 |
| DEL-END | 40.3 | 83.5 | 84.8 | 75.8 | 95.0 | 70.8 | 86.5 | 89.8 |
| $\mathbb{T}_{\text{abs}}/\mathbb{T}_{\text{rel}}$ | 40.6/68.5% | 81.6/94.3% | 84.7/96.5% | 72.4/85.7% | 94.2/97.1% | 70.2/91.2% | 85.8/95.0% | 90.4/98.5% |
| **BoolQ** | | | | | | | | |
| BASE | 75.7 | 89.3 | 88.5 | 84.2 | 90.5 | 86.2 | 88.7 | 91.3 |
| INS-BEG | 73.1 | 87.2 | 86.7 | 81.2 | 88.5 | 81.5 | 89.0 | 92.8 |
| INS-END | 74.5 | 87.5 | 86.5 | 84.8 | 89.2 | 84.0 | 91.3 | 92.0 |
| DEL-BEG | 74.8 | 88.0 | 85.3 | 82.8 | 89.5 | 82.7 | 89.2 | 92.0 |
| DEL-END | 73.3 | 89.5 | 86.7 | 81.7 | 86.8 | 83.8 | 90.0 | 92.8 |
| $\mathbb{T}_{\text{abs}}/\mathbb{T}_{\text{rel}}$ | 73.9/97.6% | 88.1/98.7% | 86.3/97.5% | 82.6/98.1% | 88.5/97.8% | 83.0/96.3% | 89.9/101.4% | 92.4/101.2% |
| **CSQA** | | | | | | | | |
| BASE | 59.7 | 69.7 | 70.7 | 66.6 | 73.7 | 67.0 | 73.5 | 75.7 |
| INS-BEG | 53.8 | 65.1 | 65.4 | 60.2 | 69.9 | 64.8 | 69.1 | 73.9 |
| INS-END | 52.0 | 67.0 | 68.5 | 56.5 | 71.9 | 62.0 | 67.6 | 73.4 |
| DEL-BEG | 46.6 | 63.0 | 60.8 | 49.3 | 65.6 | 58.9 | 63.4 | 73.3 |
| DEL-END | 50.0 | 65.2 | 65.0 | 55.8 | 69.5 | 62.0 | 66.7 | 73.9 |
| $\mathbb{T}_{\text{abs}}/\mathbb{T}_{\text{rel}}$ | 50.6/84.8% | 65.1/93.4% | 64.9/91.8% | 55.5/83.3% | 69.3/93.9% | 61.9/92.4% | 66.7/90.7% | 73.6/97.2% |

of $41.7\% \rightarrow 70.2\%$. Furthermore, the SOTA model GPT-4o (more than 80% of the blue markers) retains an average of 87.8% of its capability, whereas the weakest model, Gemma-2-2B, retains only 41.7%. This aligns with that of humans (Rayner et al., 2006; Frost, 2012) in Typoglycemia scenarios and opens up a new avenue for evaluating model capabilities (more results are in Appendix F.1.1).

**Obs.2. The degree to which LLMs' performance is affected is positively correlated with the logical complexity of the TypoC task.** In Table 1, gray markers are only seen in the BoolQ/CSQA (reasoning tasks), where LLMs retain 95.6% and 81.7% of their capabilities, respectively, compared to just 67.8% on the math (GSM8k) task, which demands more complex logical reasoning. Notably, for the yes/no BoolQ dataset, applying sentence-level $\mathcal{F}_\Omega$ results in an unusual slight average 0.7%↑ in accuracy. However, for humans, reading scrambled text typically hampers comprehension (Ferreira et al., 2002). This performance improvement may be misleadingly optimistic, suggesting that LLMs *might rely on the attention mechanism to capture certain representations from scrambled text that help derive correct results.* This statistically-driven mechanism vastly differs from the micro-level processes of human reading and understanding.

**Obs.3. The position of characters affects LLMs' understanding differently.** As shown in Table 1, the accuracy of character-level $\mathcal{F}_\Omega$ under the ALL setting is 100% lower than that of INT, while the accuracy of BEG is lower than that of END in 87.5% of the cases. This indicates that the importance of the first, last, and internal characters decreases in that order—which further reveals the similarity that both LLMs and humans pay more attention to the first and last characters (Perea & Lupker, 2004) (more results on another two datasets are placed in Appendix F.1.1).

**Obs.4. Typoglycemia leads to an increased computational cost.** As shown in Figure 3, the ratio of tokens and time before and after the $\mathcal{F}_\Omega$ transformation is greater than 1 in 100% of cases for all LLMs. For instance, GPT-3.5-Turbo exhibits a 46.5% ↑ in prompt tokens and an 18% ↑ in completion time. Similarly, (Rayner et al., 2006) finds that humans also require more eye fixations and longer fixation durations when reading Typoglycemia text. This finding reveals that both LLMs and humans struggle in Typoglycemia scenarios (more results are shown in Appendix F.3).

**Obs.5. The results of TypoP are consistent with those of TypoC.** As Table 2 shows, the performance ranking of 7 out of 8 LLMs closely mirrors that in Table 1. For example, Llama-3.1 shows the same accuracy ranking in both tables: $END > BEG > INT > ALL > REV$. This observation reveals that the robustness of LLMs in Typoglycemia scenarios is positively correlated with their ability to correct Typoglycemia text (See more results on another two TypoPs in Appendix F.2).

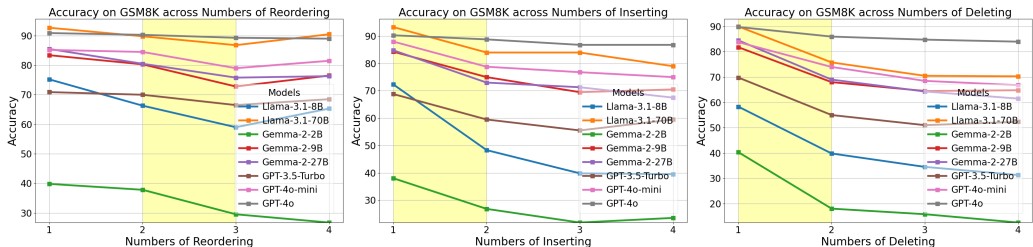

**Figure 4: The line charts of accuracy for each model**, as the number of operations increase from 1 to 4 when $\mathcal{F}_\Omega$ =REO_INT, INS_INT, and DEL_INT at *character* level on GSM8k dataset.

**Table 4: Encoder Perspective: The cosine similarity between the embedding of normal text and text processed by** $\mathcal{F}_\Omega$, using text-embedding-3 to get the vectors. **BASE** is the standard for similarity calculation.

| Datasets/$\mathcal{F}_\Omega$ | Character Level | | | | | Word Level | | | Sentence Level | | |
|---|---|---|---|---|---|---|---|---|---|---|---|
| | REO-ALL | REO-INT | REO-REV | INS-INT_3 | DEL-INT_3 | REO-ALL | REO-ADJ | REO-REV | REO-ALL | REO-ADJ | REO-REV |
| GSM8k | 0.755 | 0.891 | 0.594 | 0.865 | 0.830 | 0.930 | 0.962 | 0.915 | 0.978 | 0.987 | 0.967 |
| BoolQ | 0.912 | 0.944 | 0.863 | 0.944 | 0.933 | 0.965 | 0.978 | 0.960 | 0.986 | 0.993 | 0.980 |
| CSQA | 0.867 | 0.922 | 0.836 | 0.910 | 0.890 | 0.949 | 0.968 | 0.940 | 0.999 | 0.998 | 0.997 |

## 4.3 IMPACT OF TYPOGLYCEMIA FUNCTIONS (RQ2)

To answer RQ2, we conduct experiments using additional insertion and deletion Typoglycemia functions to verify the impact of other Typoglycemia concepts on LLMs. We list the results in Table 3 (more results are placed in Appendix F.1.2) and we can summarize the observations:

**Obs.1. The impact of Insertion and Deletion on LLMs is generally similar to Reordering, but the magnitude of the impact is smaller.** As shown in Table 3, red markers are primarily concentrated in the BASE row, and the SOTA model GPT-4o retains an average accuracy of 98.9% across the three datasets, while the weakest model, Gemma-2-2B, achieves 83.6%. In all cases, the average retained accuracy increases by $0.1 \sim 26.8\%$ compared to Reordering. This indicates that LLMs are more robust to minor additions or deletions of characters than to character reordering.

**Obs.2. Insertion and Deletion also result in an unusual increase in accuracy for tasks with weaker logic.** As shown in Table 3, 90.9% of the gray markers appear in the BoolQ dataset, which is consistent with the pattern observed in Reordering. This observation further confirms that minor perturbations in the prompt can aid models in understanding simple logical problems.

**Obs.3. LLMs exhibit sensitivity to character position for Deletion, but are less sensitive for Insertion.** As shown in Table 3, in the case of the INS operation, $BEG < END$ occurs in 54.2% of cases, whereas for the DEL operation, this ratio rises to 83.3%, a 29.1% increase. This observation reinforces the finding in RQ1 that the first letter is more important than the last, and reveals that their relative importance can vary depending on the operation.

## 4.4 SCRAMBLING RATIO OF TYPOGLYCEMIA (RQ3)

To address RQ3, we gradually increase the number of reordering, inserting, and deleting operations applied to each word's internal characters to increase the scrambling ratio of texts. The corresponding results are shown in Figure 4 and Appendix F.4, of which we derive the following observations:

**Obs.1. As the scrambling ratio increases, the TypoC task becomes more challenging for LLMs.** As shown in Figure 4, with the increasing number of operations, the accuracy of LLMs generally shows a downward trend across all three cases, with a drop ranging from 0.3% to 14.2%. We highlighted the regions with the largest decreases in yellow . This observation aligns with human behavior (Just & Carpenter, 1980), indicating that as the internal structure of the text becomes more disordered, it becomes increasingly difficult for LLMs to understand the text.

**Obs.2. Different models exhibit varying levels of resistance to scrambling text.** As shown in Figure 4, *Llama-3.1-8B demonstrates the weakest robustness, while GPT-4o shows the strongest anti-Typoglycemia ability*. The absolute values of the average slope in accuracy for the Inserting operation are 10.9 and 1.2, respectively. This robustness can serve as a measure of LLMs' ability to handle scrambled text, which may offer a new approach for evaluating LLMs.

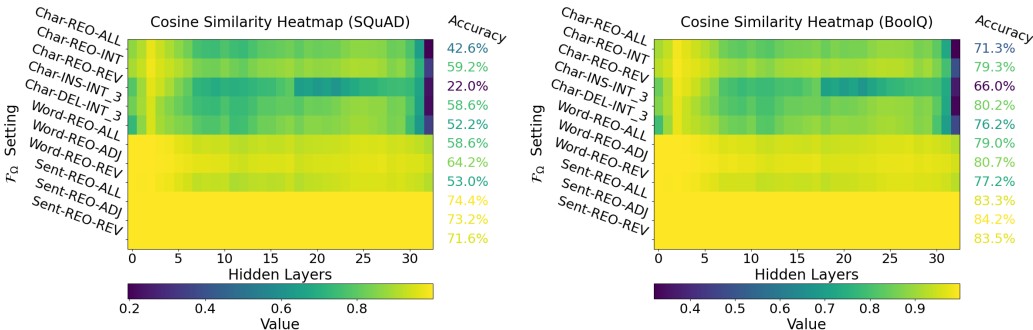

**Figure 5: Decoder Perspective: The cosine similarity between the representations of normal text and the text processed by** $\mathcal{F}_\Omega$ **in the** SQuAD **and** BoolQ **for each layer of the** Llama-3.1-8B **model (which has 33 layers in total: 1 word embedding layer and 32 Transformer layers).** **BASE** **is the standard for similarity calculation.**

### 4.5 WHY DO LLMS ALIGN WITH HUMAN PERFORMANCE (RQ4)

To answer RQ4, we embed the task text before and after being processed by $\mathcal{F}_\Omega$ using the text-embedding-3-large model and then calculate their semantic similarity. Additionally, We calculate the mean of representation tensors between Transformers, aggregating their semantics into vectors to calculate similarity with BASE. Finally, we derive the above Table 4, Figure 5 and observations:

**Obs.1. The text before and after** $\mathcal{F}_\Omega$ **processing exhibits a high degree of semantic similarity, with the impact varying depending on the level of text granularity.** As shown in Table 4, in the CSQA dataset, Typoglycemia text at the *character*, *word*, and *sentence* level retains an average semantic similarity of 0.885, 0.952, and 0.998, respectively, compared to the unprocessed text (BASE). This indicates that disturbances at the character level have the greatest impact on LLMs' understanding of the text. This observation suggests that, from the encoder's perspective, Typoglycemia text preserves a substantial amount of semantic information, which enables LLMs to exhibit robustness similar to humans in Typoglycemia scenarios (more results are placed in Appendix F.5).

**Obs.2. The subsequent representations of Typoglycemia text by LLMs are critical to their task performance.** As shown in Figure 5, the color of accuracy and representation is closely aligned. For instance, on SQuAD, for the 3 types of $\mathcal{F}_\Omega$ at the sentence level, the similarity scores of their representations are all yellow (high similarity), and corresponding accuracy is the highest at 74.4% , 73.2% , and 71.6% , respectively. In contrast, when $\mathcal{F}_\Omega$ = Char-REO-REV, the representation similarity score is the lowest (darkest color), with lowest accuracy at 22.0%. This observation demonstrates that, from the decoder's perspective, the semantic information retained in the representations across the Transformer layers is crucial for LLMs to correctly understand and respond.

**Obs.3. The hidden layer representations of the same LLM across different datasets exhibit similar "cognitive patterns."** As illustrated in Figure 5, the color distributions for SQuAD and BoolQ under various $\mathcal{F}_\Omega$ appear visually similar. Specifically, the cosine similarity between the concatenated and linearized heatmaps of these two datasets is **0.9994**, indicating a high degree of similarity. Additionally, the color distributions vary across different models when evaluated on the same dataset (See more figures in Appendix F.6). Based on these observations, we posit that the heatmap can translate *each model's unique "cognitive pattern"* through our Typoglycemia experiments, much like how different human individuals exhibit distinct cognitive patterns.

### 5 CONCLUSION

In this paper, we explore the emerging field of *LLM Psychology* by investigating the behavior of LLMs through the lens of Typoglycemia. Our study reveals how LLMs handle scrambled text, providing insights into their cognitive-like abilities and limitations. Through systematic analysis, we observe that LLMs demonstrate human-like behaviors, such as reduced task accuracy and increased token and time consumption, when faced with text distortions. Additionally, the varying robustness across different LLMs suggests that scrambled text understanding serves as an accessible benchmark for evaluating model performance. Despite some promising results, our analysis of LLMs' hidden layers reveals their reliance on data-driven mechanisms, with limited capacity for deep reasoning. By digging into the hidden layer semantics, we further reveal that **each LLM demonstrates its unique and consistent cognitive pattern** across different datasets in Typoglycemia.

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

## A  EXPERIMENTAL DISCUSSION

Through comprehensive and systematic experiments in Typoglycemia and its migrated scenarios, we discover both the **alignment** of LLMs with human cognition and their **distinct** behaviors. LLMs exhibit a decline in task accuracy, increased resource consumption, and many other human-like behaviors, such as placing greater emphasis on initial letters. This significantly advances research on aligning LLMs with human cognition and provides a solid and vivid case for our proposed "LLM Psychology." Furthermore, we observe LLMs' **counter-intuitive** and **counter-logical** performance under certain settings, offering strong evidence for the argument that LLMs possess *data-driven statistical reasoning abilities rather than human logic*. Finally, we explore the underlying causes and observe different LLM's unique cognitive pattern on these phenomena from the perspectives of *encoder* and *decoder* architectures, providing new insights into the cognitive mechanisms of LLMs.

# B DATASET DESCRIPTION

**Table 5:** Dataset Details of TypoBench

| Dataset | TypoC Task | Size | Metrics | Sample Number |
|---|---|---|---|---|
| GSM8k | Mathematical Problem Solving | 17,584 | Accuracy/CosSim | 1,200 |
| MBPP | Code Generation | 1,401 | Accuracy/CosSim | 700 |
| BoolQ | Context Question Answering (yes/no) | 12,697 | Accuracy/CosSim | 1,200 |
| SQuAD | Context Question Answering (phrases) | 98,169 | Accuracy/CosSim | 1,000 |
| CSQA | Commonsense Reasoning | 12,102 | Accuracy/CosSim | 2,000 |

## B.1 DATASET SELECTION STRATEGY

To systematically evaluate the performance of LLMs on TypoBench, we focus on three key capabilities when selecting datasets and task scenarios: **logical reasoning**, **contextual learning**, and **knowledge acquisition**.

### B.1.1 LOGICAL REASONING

**Strong Logic Tasks.** We refer to tasks that involve multi-step reasoning, where an error in one step leads to errors in subsequent steps, as Strong Logic Tasks (SLTs). Representative scenarios we select include **mathematical problem solving** (GSM8k as dataset) and **code generation** (MBPP as dataset). SLTs pose stringent challenges to the logical reasoning capabilities of LLMs. From a data-driven perspective, Typoglycemia disrupts the morpheme order in normal natural language text, which, in turn, disturbs the inherent logic, leading to confused understanding and erroneous reasoning. For humans, the combination of SLT scenarios and Typoglycemia text makes task completion nearly impossible. In a certain sense, this implies that SLTs are effective in testing LLMs' performance on TypoBench, thereby revealing their underlying cognitive mechanisms.

**Weak Logic Tasks.** Conversely, tasks with less stringent requirements for logical correctness are referred to as Weak Logic Tasks (WLTs). These tasks typically challenge LLMs' capabilities not only in simple logical reasoning but also in other areas. WLTs primarily serve as a platform for simultaneously evaluating multiple aspects of LLMs' abilities. In our experimental strategy, WLTs are combined with contextual learning and knowledge acquisition.

### B.1.2 CONTEXTUAL LEARNING

Contextual learning refers to the ability of LLMs to perceive and learn the knowledge, patterns, and other elements within the context of a given prompt. We select task datasets for contextual learning at two levels of difficulty. Given a contextual passage and a related question, LLMs are instructed to answer with either "yes/no" (BoolQ as dataset) or phrases (SQuAD as dataset), corresponding to easy and difficult settings, respectively. In the yes/no setting, the response is not directly tied to the context, allowing LLMs to rely on coarse-grained semantic understanding. However, in the phrases setting, LLMs are required to have a more localized understanding of the contextual content, posing a more severe challenge to their learning and perception capabilities. By combining these two scenarios with TypoBench, we can explore how Typoglycemia affects LLMs' ability to perceive both local and global information.

### B.1.3 KNOWLEDGE ACQUISITION

LLMs possess knowledge capabilities, which are embedded within their layer weights. Generally, a model activates and extracts the knowledge embedded in these weights through the input prompt, enabling it to generate responses. This raises an interesting question: does Typoglycemia affect this process? In our experimental strategy, we investigate whether Typoglycemia disrupts the extraction of knowledge from the model's internal weights by evaluating its impact on answering context-independent common sense questions (CSQA as dataset). This approach allows us to explore how the perturbation of input text influences the model's ability to retrieve knowledge.

## B.2 DATASET EXAMPLE

### B.2.1 GSM8K

**Question:**
Julie wants to give her favorite cousin a $2345 mountain bike for his birthday. So far, she has saved $1500. Since the birthday is still a few weeks away, Julie has time to save even more. She plans on mowing 20 lawns, delivering 600 newspapers, and walking 24 of her neighbors' dogs. She is paid $20 for each lawn, 40 cents per newspaper, and $15 per dog. After purchasing the bike, how much money will Julie have left?

**Answer:**
Mowing lawns will earn Julie $20 \times 20 = \boxed{400}$ dollars.

Her earnings, in dollars, from delivering newspapers will be $600 \times \frac{40}{100} = \boxed{240}$ dollars.

After walking 24 of her neighbor's dogs, she will earn $24 \times 15 = \boxed{360}$ dollars.

She will therefore earn a total of $400 + 240 + 360 = \boxed{1000}$ dollars.

Combining earnings from her job with her savings will give Julie $1000 + 1500 = \boxed{2500}$ dollars.

Subtracting the cost of the mountain bike from the total will leave Julie with a balance of $2500 - 2345 = \boxed{155}$ dollars.

### B.2.2 MBPP

**Text:**
Write a function to find the peak element in the given array.

**Test Cases:**
```
assert find_peak([1, 3, 20, 4, 1, 0], 6) == 2
assert find_peak([2, 3, 4, 5, 6], 5) == 4
assert find_peak([8, 9, 11, 12, 14, 15], 6) == 5
```

**Code:**
```
def find_peak_util(arr, low, high, n):
    mid = low + (high - low)/2
    mid = int(mid)
    if ((mid == 0 or arr[mid - 1] <= arr[mid]) and
        (mid == n - 1 or arr[mid + 1] <= arr[mid])):
        return mid
    elif (mid > 0 and arr[mid - 1] > arr[mid]):
        return find_peak_util(arr, low, (mid - 1), n)
    else:
        return find_peak_util(arr, (mid + 1), high, n)

def find_peak(arr, n):
    return find_peak_util(arr, 0, n - 1, n)
```

### B.2.3  BOOLQ

**Question:**
Do all bacteria have peptidoglycan in their cell walls?

**Passage:**
Peptidoglycan, also known as murein, is a polymer consisting of sugars and amino acids that forms a mesh-like layer outside the plasma membrane of most bacteria, forming the cell wall. The sugar component consists of alternating residues of $\beta$-(1,4) linked N-acetylglucosamine (NAG) and N-acetylmuramic acid (NAM). Attached to the N-acetylmuramic acid is a peptide chain of three to five amino acids. The peptide chain can be cross-linked to the peptide chain of another strand forming the 3D mesh-like layer. Peptidoglycan serves a structural role in the bacterial cell wall, giving structural strength, as well as counteracting the osmotic pressure of the cytoplasm. A common misconception is that peptidoglycan gives the cell its shape; however, whereas peptidoglycan helps maintain the structural strength of the cell, it is actually the MreB protein that facilitates cell shape. Peptidoglycan is also involved in binary fission during bacterial cell reproduction.

**Answer:**
False

### B.2.4  SQUAD

**Context:**
The control of associated biodiversity is one of the great agricultural challenges that farmers face. On monoculture farms, the approach is generally to eradicate associated diversity using a suite of biologically destructive pesticides, mechanized tools, and transgenic engineering techniques, then to rotate crops. Although some polyculture farmers use the same techniques, they also employ integrated pest management strategies as well as strategies that are more labor-intensive, but generally less dependent on capital, biotechnology, and energy.

**Question:**
What is one of the great agricultural challenges that farmers face?

**Answer:**
The control of associated biodiversity

### B.2.5  CSQA

**Question:**
John watches the well-dressed people from a catwalk above the stage. He listens to them speak rehearsed lines while the audience listens. Where is he?

**Choices:**
**A.** theatre **B.** new york city **C.** fashion show **D.** construction site **E.** school play

**Correct Answer:**
**A.** theatre

## C  TASK PROMPT

### C.1  TASK COMPLETION PROMPT

#### C.1.1  MATHEMATICAL PROBLEM SOLVING

> Solve the math problem below:
> Problem: {mathematical problem description}
> Response in the following format without any other information:
> process: <reasoning steps here>
> answer_number: <final answer number here>

#### C.1.2  MATHEMATICAL PROBLEM SOLVING

> Solve the code problem below in Python:
> Problem: {code description}
> Response in the following format without any other information:
> process: <reasoning steps here>
> code: <Python code here>

#### C.1.3  CONTEXT QUESTION ANSWERING

> Answer the question with only 'yes' or 'no' based on the passage below:
> Question: {question description}
> Passage: {context passage}
> Response in the following format without any other information:
> reason: <reason for yes or no here>
> answer: <'yes' or 'no' here>

> Answer the question with word or phrase based on the context below:
> Question: {question description}
> Passage: {context passage}
> Response in the following format without any other information:
> reason: <reason for the answer here>
> answer: <answer here>

#### C.1.4  COMMONSENSE REASONING

> Choose one choice that best answers the commonsense question below:
> Question: {question description}
> Choices: {context passage}
> Response in the following format without any other information:
> reason: <reason for the choice here>
> answer: <one choice from the choices list here>

## C.2 TASK PERCEPTION PROMPT

### C.2.1 RECTIFY

Correct the scrambled letters in each word of the following passage:
Passage: {passage text}
Response in the following format without any other information:
rectified: <rectified passage here >

## C.3 SUMMARIZE

Summarize the main content of the following passage:
Passage: {passage text}
Response in the following format without any other information:
summarized: <summarized passage here>

## C.4 TRANSLATE

Translate the following English passage into Chinese:
Passage: {passage text}
Response in the following format without any other information:
translated: <translated passage here>

# D  TYPOFUNC DESCRIPTION

Psychological experiments on Typoglycemia typically involve transposing the letters at the beginning, end, and internal positions within words. We have extended this operation to a broader set of TypoFuncs at the letter, word, and sentence levels. Additionally, at the character level, we have designed TypoFuncs such as insertion and deletion.

## D.1  CHARACTER

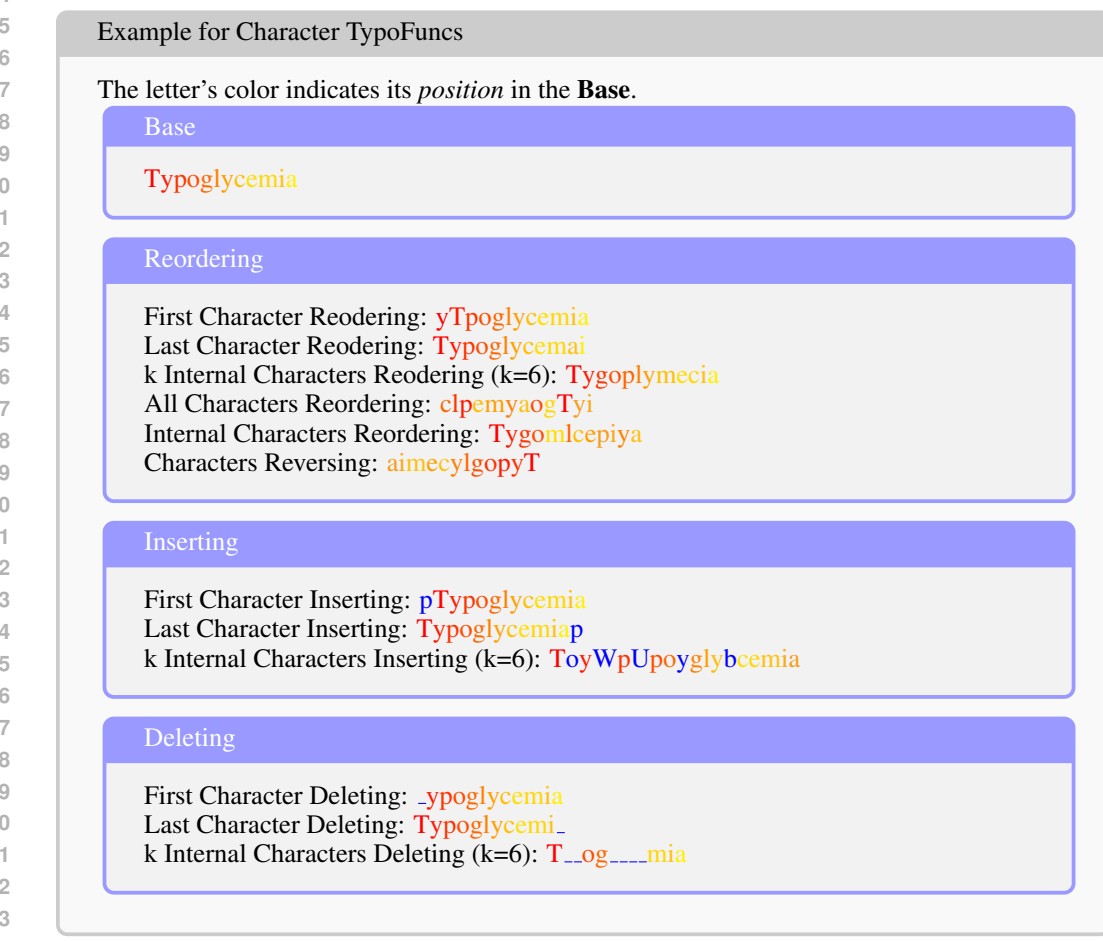

At the character level, we treat characters as the smallest operational units. TypoFuncs operate on the letters within each word. The character-level TypoFuncs are divided into three categories: *reordering*, *inserting*, and *deleting* (denoted as **R, I, D** respectively). Each of these TypoFuncs includes the following specific operations:

- **First Character R/I/D**, which performs corresponding operation on the first letter of each word.

- **Last Character R/I/D**, which performs the respective operation on the last letter of each word.

- **k Internal Characters R/I/D**, which performs the respective operation on k randomly selected internal letters (excluding the first and last) within each word.

Additionally, the reordering category includes the following specific operations:

- **All Characters Reordering**: This operation shuffles all the letters within the word.

- **Internal Characters Reordering**: This operation shuffles the letters in the middle of the word (excluding the first and last letters).

- **Characters Reversing**: This operation reverses the order of all letters within the word.

## D.2 WORD

> **Example for Word TypoFuncs**
>
> The shading of each word's color indicates its *position* in the **Base**.
>
> | Base | All Words Reordering |
> | --- | --- |
> | Julie wants to give her cousin a $2345 mountain bike for his birthday. | $2345 a to bike birthday cousin give her Julie wants mountain his for. |
>
> | Adjacent Words Reordering | Words Reversing |
> | --- | --- |
> | Julie to wants her give a cousin $2345 mountain bike his for birthday. | birthday his for bike mountain $2345 a cousin her give to wants Julie. |

On the word level, we consider words to be the basic operational units. Given the substantial effect of inserting and deleting words on the overall meaning, which can cause either nuanced or substantial semantic redundancy or loss, our primary emphasis is on the following reordering operations:

- **All Words Reordering**, which randomly shuffles the words within each sentence.
- **Adjacent Words Reordering**, which randomly swaps adjacent words within each sentence.
- **Words Reversing**, which reverses the order of words within each sentence.

## D.3 SENTENCE

> **Example for Sentence TypoFuncs**
>
> The shading of each sentence's color indicates its *position* in the **Base**.
>
> | Base | All Words Reordering |
> | --- | --- |
> | The sun rises early every morning. Birds sing softly in the trees. Flowers bloom in vibrant colors daily. Children play happily in the park. People walk briskly to their jobs. Evening arrives with a peaceful calm. | Children play happily in the park. Birds sing softly in the trees. Flowers bloom in vibrant colors daily. People walk briskly to their jobs. The sun rises early every morning. Evening arrives with a peaceful calm. |
>
> | Adjacent Words Reordering | Words Reversing |
> | --- | --- |
> | The sun rises early every morning. Flowers bloom in vibrant colors daily. Birds sing softly in the trees. People walk briskly to their jobs. Children play happily in the park. Evening arrives with a peaceful calm. | Evening arrives with a peaceful calm. People walk briskly to their jobs. Children play happily in the park. Flowers bloom in vibrant colors daily. Birds sing softly in the trees. The sun rises early every morning. |

On the sentence level, sentences are regarded as the basic operational units. Likewise, because of the considerable influence that sentence insertion and deletion have on textual meaning, our main focus is on the reordering operations detailed below:

- **All Sentences Reordering**, which randomly shuffles the sentences within the text.
- **Adjacent Sentences Reordering**, which randomly swaps adjacent sentences within the text.
- **Sentences Reversing**, which reverses the order of sentences within the text.

## E    PARAMETER SETTINGS

To ensure stability and consistency in the model outputs, we set $top\_p = 1$, $n = 1$, $frequency\_penalty = 0$, and $presence\_penalty = 0$ for all models. The temperature is set to 0 for GPT series models, and to $10^{-6}$ for Llama and Gemma series models.

## F    MORE RESULTS

### F.1    TYPOC

In this subsection, we further present the performance of various LLMs on two additional datasets, MBPP and SQuAD, in the TypoC task. The conclusions drawn from these results are consistent with those in the main text, further supporting the findings on the impact of the Typoglycemia scenario on LLMs.

#### F.1.1    REORDERING

**Table 6: Results on the TypoC tasks when $\mathcal{F}_\Omega = $ REO at *character*, *word*, and *sentence* levels**. We evaluate the average task accuracy (over 3 runs) of various LLMs on the MBPP and SQuAD datasets. **BASE** refers to the scenario where $\mathcal{F}_\Omega$ is not applied. In each dataset, red (blue) marks the maximum value in each row (column), and green marks values that are the maximum in both. Gray marks the values that are higher than BASE in each row. MBPP and SQuAD report the cosine similarity and accuracy, respectively. **BASE** of GPT-4o is the standard for similarity calculation. Gemma-2 series and Llama-3.1-8B fail to generate required format of code on MBPP dataset (See TypoC cases in Appendix G)

| Models/Datasets | Standard | Character · | | | | | Word | | | Sentence | | |
|---|---|---|---|---|---|---|---|---|---|---|---|---|
| | BASE | ALL | INT | BEG | END | REV | ALL | ADJ | REV | ALL | ADJ | REV |
| **MBPP** | | | | | | | | | | | | |
| Llama-3.1-70B | 0.776 | 0.468 | 0.702 | 0.782 | 0.784 | 0.267 | 0.722 | 0.753 | 0.723 | 0.774 | 0.775 | 0.778 |
| GPT-3.5-Turbo | 0.785 | 0.460 | 0.719 | 0.769 | 0.764 | 0.665 | 0.680 | 0.724 | 0.665 | 0.784 | 0.784 | 0.786 |
| GPT-4o-mini | 0.899 | 0.611 | 0.826 | 0.870 | 0.882 | 0.735 | 0.793 | 0.841 | 0.799 | 0.897 | 0.897 | 0.896 |
| GPT-4o | 1.00 | 0.766 | 0.895 | 0.924 | 0.941 | 0.902 | 0.857 | 0.897 | 0.853 | 0.982 | 0.984 | 0.962 |
| **SQuAD** | | | | | | | | | | | | |
| Gemma-2-2B | 73.8 | 27.4 | 48.8 | 54.2 | 54.4 | 8.0 | 52.8 | 57.5 | 46.8 | 72.0 | 75.2 | 70.6 |
| Gemma-2-9B | 81.4 | 52.8 | 70.0 | 76.6 | 71.0 | 24.2 | 79.2 | 66.8 | 65.0 | 81.0 | 80.6 | 79.2 |
| Gemma-2-27B | 84.6 | 83.4 | 84.2 | 78.8 | 76.6 | 29.4 | 67.6 | 74.2 | 62.0 | 83.4 | 84.2 | 83.8 |
| Llama-3.1-8B | 73.0 | 42.6 | 59.2 | 62.8 | 64.5 | 22.0 | 58.6 | 64.2 | 53.0 | 74.4 | 73.2 | 71.6 |
| Llama-3.1-70B | 84.4 | 63.0 | 75.4 | 79..2 | 79.0 | 32.8 | 70.8 | 74.6 | 65.8 | 83.0 | 83.6 | 82.2 |
| GPT-3.5-Turbo | 77.8 | 55.4 | 67.2 | 77.0 | 72.2 | 33.6 | 59.4 | 65.6 | 54.4 | 75.8 | 76.8 | 76.2 |
| GPT-4o-mini | 82.8 | 54.8 | 69.8 | 75.4 | 76.0 | 46.6 | 65.4 | 71.8 | 62.8 | 81.0 | 81.2 | 80.4 |
| GPT-4o | 88.0 | 78.2 | 79.0 | 82.8 | 81.0 | 77.6 | 73.8 | 79.4 | 70.0 | 86.8 | 86.0 | 86.6 |

### F.1.2 INSERTION AND DELETION

**Table 7: Results on the TypoC tasks when $\mathcal{F}_\Omega =$ INS and DEL at *character* level.** We evaluate the average task accuracy (over 3 runs) of various LLMs on the MBPP and SQuAD datasets. **BASE** refers to the scenario where $\mathcal{F}_\Omega$ is not applied. MBPP and SQuAD report the cosine similarity and accuracy, respectively. **BASE** of GPT-4o is the standard for similarity calculation. In each dataset, red marks the maximum value in each column. Gray marks the values that are higher than BASE in columns. Gemma-2 series and Llama-3.1-8B fail to generate required format of code on MBPP dataset (See TypoC cases in Appendix G)

| Datasets/$\mathcal{F}_\Omega$ | Gemma-2-2B | Gemma-2-9B | Gemma-2-27B | Llama-3.1-8B | Llama-3.1-70B | GPT-3.5-Turbo | GPT-4o-mini | GPT-4o |
|---|---|---|---|---|---|---|---|---|
| **MBPP** | | | | | | | | |
| BASE | – | – | – | – | 0.776 | 0.785 | 0.899 | 1.00 |
| INS-BEG | – | – | – | – | 0.740 | 0.749 | 0.858 | 0.928 |
| INS-END | – | – | – | – | 0.662 | 0.765 | 0.881 | 0.937 |
| DEL-BEG | – | – | – | – | 0.816 | 0.748 | 0.873 | 0.915 |
| DEL-END | – | – | – | – | 0.825 | 0.751 | 0.877 | 0.935 |
| $\mathbb{T}_{abs}/\mathbb{T}_{rel}$ | – | – | – | – | 0.761/98.1% | 0.753/95.9% | 0.873/97.1% | 0.929/92.9% |
| **SQuAD** | | | | | | | | |
| BASE | 73.8 | 81.4 | 84.6 | 73.0 | 84.4 | 77.8 | 82.8 | 88.0 |
| INS-BEG | 64.6 | 79.8 | 82.4 | 73.4 | 84.6 | 77.0 | 82.0 | 85.6 |
| INS-END | 57.6 | 74.2 | 81.6 | 68.6 | 83.8 | 72.2 | 76.2 | 83.6 |
| DEL-BEG | 63.2 | 77.6 | 81.4 | 65.4 | 84.2 | 71.4 | 78.6 | 85.0 |
| DEL-END | 55.4 | 74.6 | 79.4 | 63.4 | 77.4 | 70.0 | 75.2 | 85.2 |
| $\mathbb{T}_{abs}/\mathbb{T}_{rel}$ | 60.2/81.6% | 76.5/94.0% | 81.2/96.0% | 66.7/92.7% | 82.5/97.7% | 72.7/93.4% | 78.0/94.2% | 84.9/96.5% |

### F.2 TYPOP

In this subsection, we present the performance of various LLMs on two additional tasks in TypoP: Summarize and Translate. The conclusions drawn from these results are consistent with those in the main text: the results of TypoP align with those of TypoC.

**Table 8: Results (Cosine Similarity) on TypoP-Summarize and Translate tasks** when $\mathcal{F}_\Omega$ is set to **REO** on *character* level for BoolQ dataset. **BASE** is the standard for similarity calculation. In each TypoTask, red (blue) marks the maximum value in each row (column), and green marks values that are the maximum in both (See TypoP cases in Appendix H)

| Models/Tasks | REO | | | | | INS | | | DEL | | |
|---|---|---|---|---|---|---|---|---|---|---|---|
| | ALL | INT | BEG | END | REV | BEG | INT_1 | END | BEG | INT_1 | END |
| **Summarize** | | | | | | | | | | | |
| Gemma-2-2B | 0.421 | 0.777 | 0.821 | 0.861 | 0.119 | 0.886 | 0.871 | 0.875 | 0.845 | 0.876 | 0.866 |
| Gemma-2-9B | 0.633 | 0.859 | 0.898 | 0.908 | 0.218 | 0.918 | 0.915 | 0.916 | 0.908 | 0.923 | 0.914 |
| Gemma-2-27B | 0.624 | 0.867 | 0.901 | 0.912 | 0.273 | 0.923 | 0.921 | 0.939 | 0.907 | 0.921 | 0.917 |
| Llama-3.1-8B | 0.500 | 0.778 | 0.849 | 0.842 | 0.147 | 0.901 | 0.881 | 0.876 | 0.855 | 0.841 | 0.868 |
| Llama-3.1-70B | 0.684 | 0.869 | 0.915 | 0.922 | 0.274 | 0.933 | 0.926 | 0.924 | 0.918 | 0.925 | 0.926 |
| GPT-3.5-Turbo | 0.687 | 0.867 | 0.915 | 0.917 | 0.463 | 0.916 | 0.924 | 0.922 | 0.909 | 0.916 | 0.912 |
| GPT-4o-mini | 0.695 | 0.881 | 0.926 | 0.928 | 0.772 | 0.942 | 0.935 | 0.937 | 0.934 | 0.941 | 0.937 |
| GPT-4o | 0.889 | 0.926 | 0.945 | 0.946 | 0.936 | 0.943 | 0.935 | 0.940 | 0.936 | 0.943 | 0.941 |
| **Translate** | | | | | | | | | | | |
| Gemma-2-2B | 0.229 | 0.367 | 0.402 | 0.465 | 0.142 | | | | | | |
| Gemma-2-9B | 0.489 | 0.823 | 0.876 | 0.902 | 0.153 | | | | | | |
| Gemma-2-27B | 0.496 | 0.741 | 0.779 | 0.796 | 0.234 | | | | | | |
| Llama-3.1-8B | 0.412 | 0.674 | 0.766 | 0.803 | 0.162 | | | | | | |
| Llama-3.1-70B | 0.612 | 0.832 | 0.892 | 0.902 | 0.321 | | | | | | |
| GPT-3.5-Turbo | 0.569 | 0.827 | 0.902 | 0.917 | 0.299 | | | | | | |
| GPT-4o-mini | 0.636 | 0.866 | 0.920 | 0.935 | 0.728 | | | | | | |
| GPT-4o | 0.783 | 0.883 | 0.907 | 0.924 | 0.872 | | | | | | |

### F.3 COMPLETION TIME AND PROMPT TOKENS

In this subsection, we present additional results on completion time and prompt token usage. First, we use the different datasets to comprehensively evaluate token and time consumption across different levels of scrambled input. The findings align with the conclusions drawn in the main text, further demonstrating strong parallels with human performance in scrambled reading scenarios.

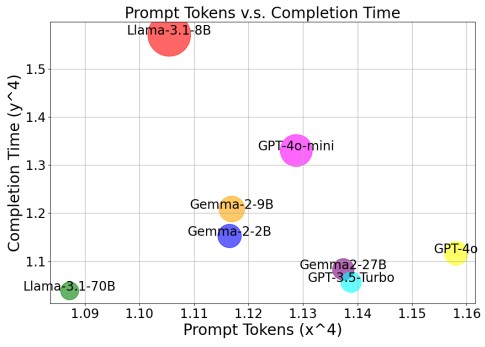

Figure 6: **Consumption ratio** before and after being processed by TypoFunc when $\mathcal{F}_\Omega$ is set to REO-ALL on character level for CSQA.

Figure 7: **Consumption ratio** before and after being processed by TypoFunc when $\mathcal{F}_\Omega$ is set to REO-INT on character level for CSQA.

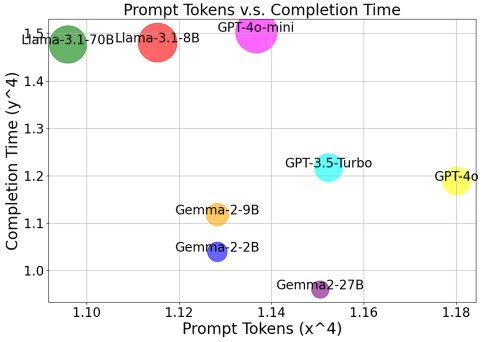
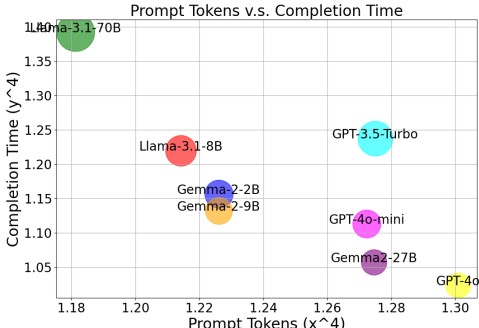

Figure 8: **Consumption ratio** before and after being processed by TypoFunc when $\mathcal{F}_\Omega$ is set to REO-REV on character level for CSQA.

Figure 9: **Consumption ratio** before and after being processed by TypoFunc when $\mathcal{F}_\Omega$ is set to ADD-BEG on character level for GSM8k.

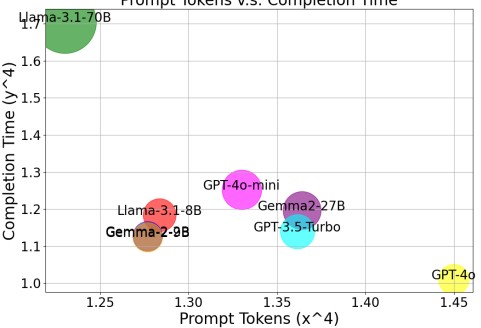
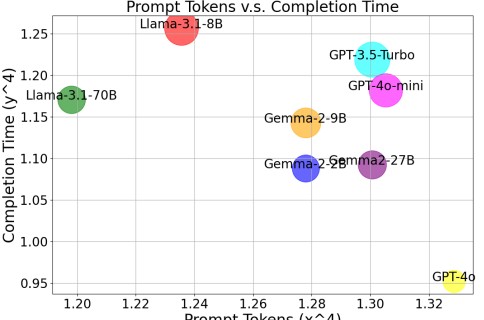

Figure 10: **Consumption ratio** before and after being processed by TypoFunc when $\mathcal{F}_\Omega$ is set to ADD-END on character level for GSM8k.

Figure 11: **Consumption ratio** before and after being processed by TypoFunc when $\mathcal{F}_\Omega$ is set to DEL-BEG on character level for GSM8k.

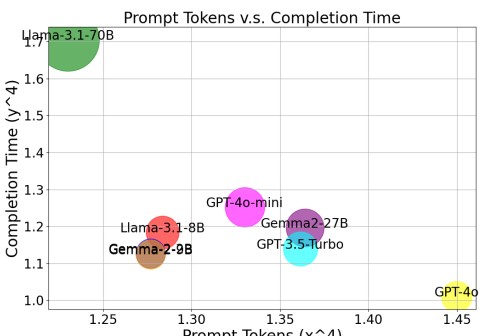

**Figure 12: Consumption ratio** when before and after being processed by TypoFunc $\mathcal{F}_\Omega$ is set to DEL-END on character level for GSM8k.

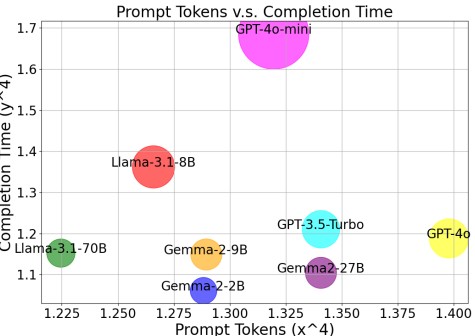

**Figure 13: Consumption ratio** when before and after being processed by TypoFunc $\mathcal{F}_\Omega$ is set to REO-INT_3 on character level for GSM8k.

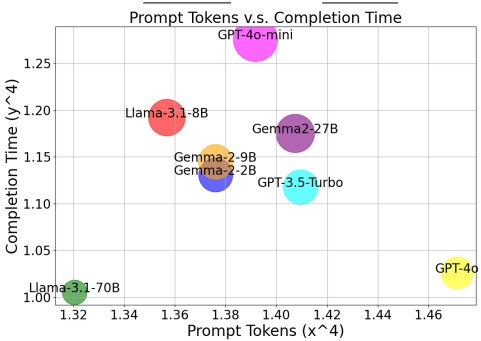

**Figure 14: Consumption ratio** when before and after being processed by TypoFunc $\mathcal{F}_\Omega$ is set to REO-BEG on character level for BoolQ.

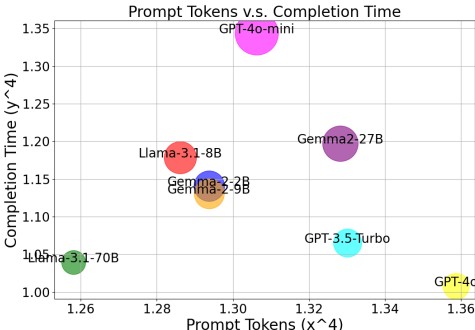

**Figure 15: Consumption ratio** when before and after being processed by TypoFunc $\mathcal{F}_\Omega$ is set to REO-END on character level for BoolQ.

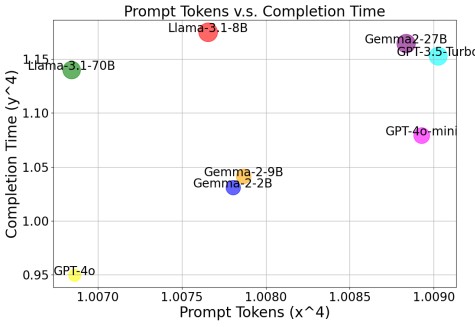

**Figure 16: Consumption ratio** when before and after being processed by TypoFunc $\mathcal{F}_\Omega$ is set to REO-ALL on word level for BoolQ.

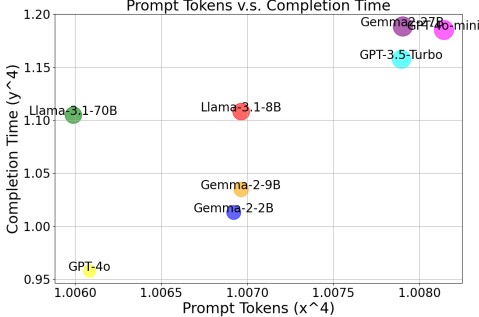

**Figure 17: Consumption ratio** when before and after being processed by TypoFunc $\mathcal{F}_\Omega$ is set to REO-ADJ on word level for BoolQ.

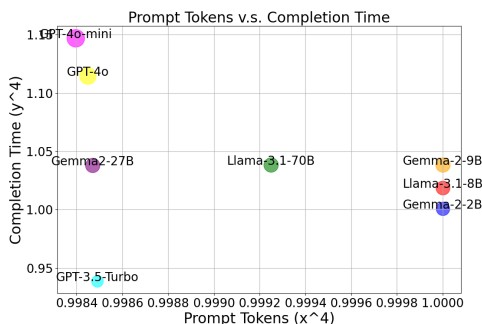

**Figure 18: Consumption ratio** when before and after being processed by TypoFunc $\mathcal{F}_\Omega$ is set to REO-ADJ on *sentence* level for MBPP.

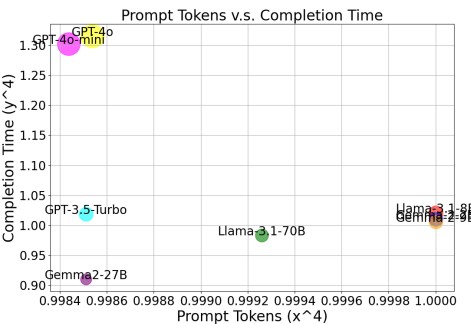

**Figure 19: Consumption ratio** when before and after being processed by TypoFunc $\mathcal{F}_\Omega$ is set to REO-REV on *sentence* level for MBPP.

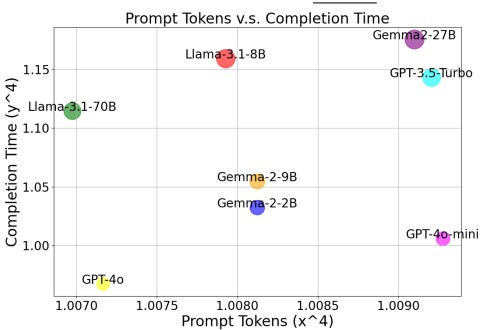

**Figure 20: Consumption ratio** when before and after being processed by TypoFunc $\mathcal{F}_\Omega$ is set to REO-REV on *word* level for BoolQ.

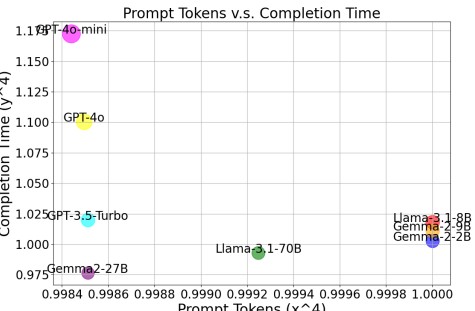

**Figure 21: Consumption ratio** when before and after being processed by TypoFunc $\mathcal{F}_\Omega$ is set to REO-ALL on *sentence* level for MBPP.

## F.4 SCRAMBLING RATIO

In this subsection, we provide charts illustrating the number of Reordering, Inserting, and Deleting operations in various task scenarios across additional datasets, along with LLMs' task performance. The results shown in these charts are similar to the findings in the main text, further validating the impact of the scrambling ratio on LLMs.

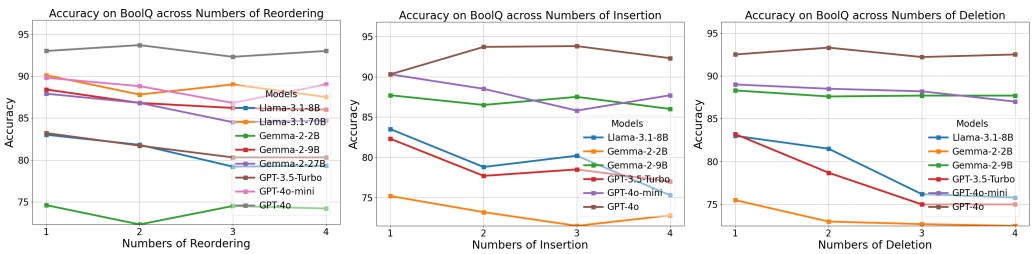

**Figure 22: The line charts of accuracy for each model**, as the number of operations increase from 1 to 4 when $\mathcal{F}_\Omega$ =REO_INT, INS_INT, and DEL_INT on BoolQ datset.

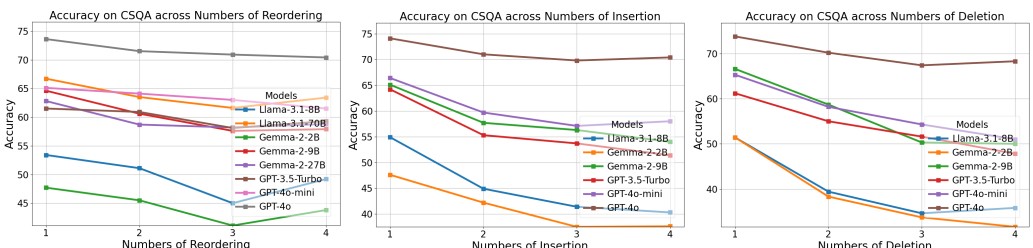

**Figure 23: The line charts of accuracy for each model**, as the number of operations increase from 1 to 4 when $\mathcal{F}_\Omega$ =REO_INT, INS_INT, and DEL_INT on CSQA datset.

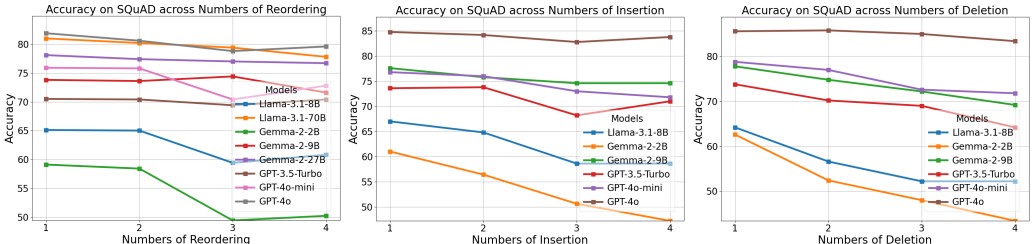

**Figure 24: The line charts of accuracy for each model**, as the number of operations increase from 1 to 4 when $\mathcal{F}_\Omega$ =REO_INT, INS_INT, and DEL_INT on SQuAD datset.

## F.5 ENCODER EMBEDDING

In this subsection, we present the similarity of text embedding across additional datasets and the similarity of input text representations at each layer of the Transformer across more models and datasets. The results shown in these charts are consistent with those in the main text, providing further data to support the related conclusions.

**Table 9: The cosine similarity between the embedding of normal text and text processed by $\mathcal{F}_\Omega$, using text-embedding-3 to get the vectors. BASE is the standard for similarity calculation.**

| $\mathcal{F}_\Omega$/Datasets | MBPP | SQuAD |
|---|---|---|
| Char-REO-ALL | 0.613 | 0.857 |
| Char-REO-INT | 0.785 | 0.912 |
| Char-REO-REV | 0.513 | 0.788 |
| Char-REO-BEG | 0.861 | 0.943 |
| Char-REO-END | 0.903 | 0.946 |
| Char-INS-BEG | 0.808 | 0.947 |
| Char-INS-END | 0.899 | 0.947 |
| Char-INS-INT_1 | 0.900 | 0.952 |
| Char-INS-INT_2 | 0.821 | 0.929 |
| Char-INS-INT_3 | 0.784 | 0.916 |
| Char-DEL-BEG | 0.874 | 0.930 |
| Char-DEL-END | 0.886 | 0.934 |
| Char-DEL-INT_1 | 0.882 | 0.950 |
| Char-DEL-INT_2 | 0.778 | 0.914 |
| Char-DEL-INT_3 | 0.727 | 0.893 |
| Word-REO-ALL | 0.845 | 0.957 |
| Word-REO-ADJ | 0.896 | 0.973 |
| Word-REO-REV | 0.815 | 0.950 |
| Sent-REO-ALL | 0.998 | 0.980 |
| Sent-REO-ADJ | 0.998 | 0.991 |
| Sent-REO-REV | 0.998 | 0.971 |

## F.6 DECODER REPRESENTATION

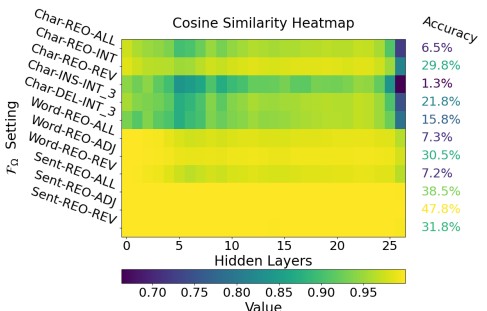
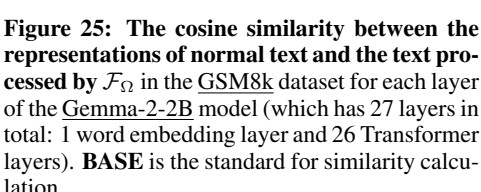
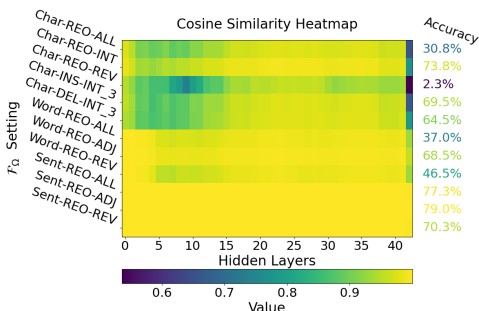

**Figure 25: The cosine similarity between the representations of normal text and the text processed by $\mathcal{F}_\Omega$ in the GSM8k dataset for each layer of the Gemma-2-2B model (which has 27 layers in total: 1 word embedding layer and 26 Transformer layers). BASE is the standard for similarity calculation.**

**Figure 26: The cosine similarity between the representations of normal text and the text processed by $\mathcal{F}_\Omega$ in the GSM8k dataset for each layer of the Gemma-2-9B model (which has 43 layers in total: 1 word embedding layer and 42 Transformer layers). BASE is the standard for similarity calculation.**

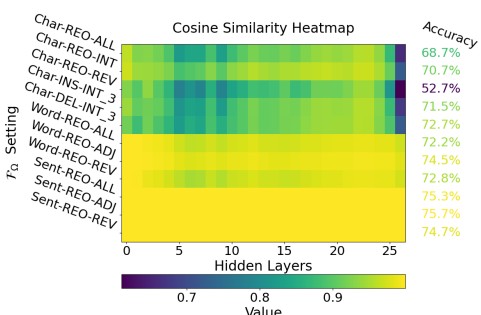

**Figure 27: The cosine similarity between the representations of normal text and the text processed by $\mathcal{F}_\Omega$ in the BoolQ dataset for each layer of the Gemma-2-2B model (which has 27 layers in total: 1 word embedding layer and 26 Transformer layers). BASE is the standard for similarity calculation.**

**Figure 28: The cosine similarity between the representations of normal text and the text processed by $\mathcal{F}_\Omega$ in the BoolQ dataset for each layer of the Gemma-2-9B model (which has 43 layers in total: 1 word embedding layer and 42 Transformer layers). BASE is the standard for similarity calculation.**

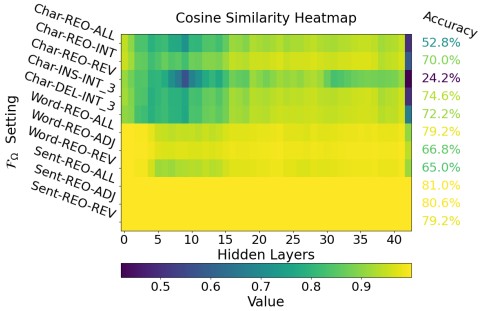
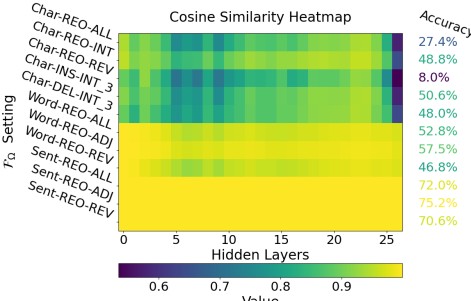

**Figure 29: The cosine similarity between the representations of normal text and the text processed by $\mathcal{F}_\Omega$ in the SQuAD dataset for each layer of the Gemma-2-9B model (which has 43 layers in total: 1 word embedding layer and 42 Transformer layers). BASE is the standard for similarity calculation.**

**Figure 30: The cosine similarity between the representations of normal text and the text processed by $\mathcal{F}_\Omega$ in the SQuAD dataset for each layer of the Gemma-2-2B model (which has 27 layers in total: 1 word embedding layer and 26 Transformer layers). BASE is the standard for similarity calculation.**

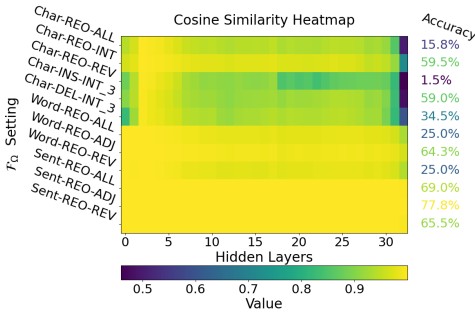

**Figure 31: The cosine similarity between the representations of normal text and the text processed by $\mathcal{F}_\Omega$ in the GSM8k dataset for each layer of the Llama-3.1-8B model (which has 33 layers in total: 1 word embedding layer and 32 Transformer layers). BASE is the standard for similarity calculation.**

# G TYPOC CASES

## G.1 MATHEMATICAL PROBLEM SOLVING (GSM8K)

### G.1.1 GOOD MIND

One case of GPT-4o's correct responses when $\mathcal{F}_\Omega =$**Char-REO-INT**.

**Typoglycemia Prompt:**
Solve the math problem below:
**Problem**: Jluie wtnas to gvie her firoatve cusoin a \$2345 mtinauon bkie for his btrihday. So far, she has saved \$1500. Sncie the bdirthay is slitl a few wekes away, Jilue has time to save eevn more. She palns on miwong 20 lawns, dienliervg 600 npawsreeps, and wianklg 24 of her nhbriegos' dgos. She is paid \$20 for each lawn, 40 cnets per nepawespr, and \$15 per dog. Afetr puhsicanrg the bike, how much menoy wlil Jilue have left?
Response in the following format without any other information:
**process**: {reasoning steps here}
**answer_number**: {final answer number here}

**Original Question:**
Julie wants to give her favorite cousin a \$2345 mountain bike for his birthday. So far, she has saved \$1500. Since the birthday is still a few weeks away, Julie has time to save even more. She plans on mowing 20 lawns, delivering 600 newspapers, and walking 24 of her neighbors' dogs. She is paid \$20 for each lawn, 40 cents per newspaper, and \$15 per dog. After purchasing the bike, how much money will Julie have left?

**Response from GPT-4o:**
**process**: 1. Calculate the total amount Julie will earn from mowing lawns: 20 lawns * \$20/lawn = \$400.
2. Calculate the total amount Julie will earn from delivering newspapers: 600 newspapers * \$0.40/newspaper = \$240.
3. Calculate the total amount Julie will earn from walking dogs: 24 dogs * \$15/dog = \$360.
4. Add the amounts from steps 1, 2, and 3 to find the total additional money Julie will earn: \$400 + \$240 + \$360 = \$1000.
5. Add the amount Julie has already saved to the total additional money: \$1500 + \$1000 = \$2500.
6. Subtract the cost of the bike from the total money Julie will have: \$2500 - \$2345 = \$155.
### 155
Let Sarah have $x$ cents. Then, Bigrde has $x + 50$ cents. Together, they have $x+(x+50) = 300$ cents (since \$3 is 300 cents). Simplifying, we get $2x + 50 = 300$. Subtract 50 from both sides to get $2x = 250$. Divide both sides by 2 to get $x = 125$.
**answer_number**: 155

**Answer from Dataset:**
Mowing lawns will earn Julie 20 * 20 = $\langle\langle 20 * 20 = 400\rangle\rangle$ 400 dollars
Her earnings, in dollars, from delivering newspapers will be 600 * 40 / 100 = $\langle\langle 600 * 40/100 = 240\rangle\rangle$ 240 dollars
After walking 24 of her neighbor's dogs, she will earn 24 * 15 = $\langle\langle 24*15 = 360\rangle\rangle$ 360 dollars.
She will therefore earn a total of 400 + 240 + 360 = $\langle\langle 400+240+360 = 1000\rangle\rangle$ 1000 dollars.
Combining earnings from her job with her savings will give Julie 1000 + 1500 = $\langle\langle 1000+1500 = 2500\rangle\rangle$ 2500 dollars.
Subtracting the cost of the mountain bike from the total will leave Julie with a balance of 2500 - 2345 = $\langle\langle 2500 - 2345 = 155\rangle\rangle$ 155 dollars.
#### 155

### G.1.2 BAD MIND

One case of GPT-4o's incorrect responses when $\mathcal{F}_\Omega =$ **Char-REO-ALL**.

**Typoglycemia Prompt:**
Solve the math problem below:
**Problem**: iamrSa is the nassatsti oachc of a rccseo aetm gynlpai saingta one of the steb temsa in hreit ueealg. She has rfuo zneod etrwa blsotet dlefli thwi awrte in a box. In the itsfr kbrae of the chmta, the 11 apelsry on the fdlei ceah teka two tltoesb of rtawe from iamaSr's box, and at the end of the eagm, kate one eorm tetbol chea. How ymna eosbttl of erwat are miinrange in Saimar's box?
**process**: {reasoning steps here}
**answer_number**: {final answer number here}

**Original Quetsion:**
Samira is the assistant coach of a soccer team playing against one of the best teams in their league. She has four dozen water bottles filled with water in a box. In the first break of the match, the 11 players on the field each take two bottles of water from Samira's box, and at the end of the game, take one more bottle each. How many bottles of water are remaining in Samira's box?

**Response from GPT-4o:**
**process**: 1. Start with 4 bottles of water.
2. Each of the 11 players takes 2 bottles during the first break: 11 players * 2 bottles = 22 bottles.
3. Each of the 11 players takes 1 more bottle at the end of the game: 11 players * 1 bottle = 11 bottles.
4. Total bottles taken: 22 bottles + 11 bottles = 33 bottles.
5. Since there were only 4 bottles initially, and 33 bottles were taken, it implies that the problem is not feasible as described. Therefore, the remaining bottles would be 0.
**answer_number**: 0

**Answer from Dataset:**
If the box has four dozen bottles of water, there are 4 * 12 = $\langle\langle 4 * 12 = 48 \rangle\rangle$ 48 bottles of water in the box
After the first half, the 11 players take 11 * 2 = $\langle\langle 11 * 2 = 22 \rangle\rangle$ 22 bottles of water from the box.
If they take 11 more bottles of water at the end of the game, the number increases to 22 + 11 = 33 bottles of water taken.
Without the bottles taken, there are 48 - 33 = $\langle\langle 48 - 33 = 15 \rangle\rangle$ 15 bottles of water remaining in the box.
#### 15

## G.2 CONTEXT QUESTION ANSWERING WITH YES/NO (BOOLQ)

### G.2.1 GOOD MIND

One case of Gemma-27B's correct responses when $\mathcal{F}_\Omega =$ **Word-REO-ALL**.

**Typoglycemia Prompt:**
Answer the question with only 'yes' or 'no' based on the passage below:
**Question**: is it possible to have a score of 1 in football
**Passage**: rare not common American method occurrences since of are – football are but the 1932 scoring in Safeties least, has every a occurred 14 safety once. Football in (NFL) the games 31 League National, under scheduling about week or rules current a once. On 21 October, 1973, Fred Dryer, Rams Bay against the Angeles Packers Los for Green the playing, score only NFL game safeties to History a in single became the in player two. A occurrence is one-point safety rarer much the, offense which by two-point the on attempt scored an be extra can conversion point or; least in NCAA have Division occurred I since 1996 at those twice football, most Bowl at 2013 the Fiesta recently. in No conversion at 1940 least since the have NFL occurred safeties. by the possible conversion defense is A also safety, though highly unlikely; never has this although occurred, a finish an point team only the single way possible football game is a with it could American in.
Response in the following format without any other information:
**reason**: {reason for yes or no here}
**answer**: {'yes' or 'no' here}

**Original Question:**
**question**: is it possible to have a score of 1 in football
**passage**: Safeties are the least common method of scoring in American football but are not rare occurrences – since 1932, a safety has occurred once every 14.31 games in the National Football League (NFL), or about once a week under current scheduling rules. On October 21, 1973, Fred Dryer, playing for the Los Angeles Rams against the Green Bay Packers, became the only player in NFL History to score two safeties in a single game. A much rarer occurrence is the one-point safety, which can be scored by the offense on an extra point or two-point conversion attempt; those have occurred at least twice in NCAA Division I football since 1996, most recently at the 2013 Fiesta Bowl. No conversion safeties have occurred since at least 1940 in the NFL. A conversion safety by the defense is also possible, though highly unlikely; although this has never occurred, it is the only possible way a team could finish with a single point in an American football game.

**Answer from Dataset:**
True

**Response from Gemma-27B:**
**reason**: The passage states that it is possible for a team to finish a game with only one point.
**answer**: yes

### G.2.2 BAD MIND

One case of Gemma-27B's incorrect responses when $\mathcal{F}_\Omega =$**Word-REO-REV**.

**Typoglycemia Prompt:**
Answer the question with only 'yes' or 'no' based on the passage below:
**Question**: is there a train station at burnham on sea
**Passage**: closure station's After, demolished were shed goods and platform the. Drive Marine into developed was eastwards trackbed former The, town the into flow traffic ease helped has which road a. Street Abingdon and Approach Station Old of junction the on is itself station the of location The, park car small a now is yard goods former the and. Dorset & Somerset the renamed subsequently was Street High the on pub adjacent An. building station main former the of east and south the to land occupy yard and station RNLI new The, platform excursion former the of site the to adjacent. side seaward-west the to built was supermarket Somerfield specifically-designed ever first The, Morrisons to sold then was which. Response in the following format without any other information:
**reason**: {reason for yes or no here}
**answer**: {'yes' or 'no' here}

**Response from Gemma-27B:**
**reason**: The passage states that "itself station the of location The, park car small a now is yard goods former the and. **answer**: yes

**Original Quetsion:**
**question**: is there a train station at burnham on sea
**passage**: After station's closure, the platform and goods shed were demolished. The former trackbed eastwards was developed into Marine Drive, a road which has helped ease traffic flow into the town. The location of the station itself is on the junction of Old Station Approach and Abingdon Street, and the former goods yard is now a small car park. An adjacent pub on the High Street was subsequently renamed the Somerset & Dorset. The new RNLI station and yard occupy land to the south and east of the former main station building, adjacent to the site of the former excursion platform. The first ever specifically-designed Somerfield supermarket was built to the seaward-west side, which was then sold to Morrisons.

**Answer from Dataset:**
False

## G.3 Context Question Answering with phrases (SQuAD)

### G.3.1 Good Mind

One case of Llama-3.1-70B's correct responses when $\mathcal{F}_\Omega =$ **Char-DEL-BEG**.

**Typoglycemia Prompt:**
Answer the question with word or phrase based on the context below:
**Question**: What was the trade name of amphetamine as a nasal decongestant?
**Context**: _hile _ighly _ffective, the _equirement for _njection _imited the use of _orepinephrine[_larification _eeded] and _rally _ctive _erivatives _ere _ought. A _tructurally _imilar _ompound, _phedrine, was _dentified by _apanese _hemists in the Ma _uang _lant and _arketed by Eli _illy as an _ral _reatment for _sthma. _ollowing the _ork of _enry _ale and _eorge _arger at _urroughs-_ellcome, _cademic _hemist _ordon _lles _ynthesized _mphetamine and _ested it in _sthma _atients in 1929. The _rug _roved to _ave _nly _odest _nti-_sthma _ffects, but _roduced _ensations of _xhilaration and _alpitations. _mphetamine was _eveloped by _mith, _line and _rench as a _asal _econgestant _nder the _rade _ame _enzedrine _nhaler. _mphetamine was _ventually _eveloped for the _reatment of _arcolepsy, _ost-_ncepheletic _arkinsonism, and _ood _levation in _epression and _ther _sychiatric _ndications. It _eceived _pproval as a New and _onofficial _emedy _rom the _merican _edical _ssociation for _hese _ses in 1937 and _emained in _ommon use for _epression _ntil the _evelopment of _ricyclic _ntidepressants in the1960s.
Response in the following format without any other information:
**reason**: {reason for the answer here}
**answer**: {answer here}

**Original Question:**
**context**: While highly effective, the requirement for injection limited the use of norepinephrine[clarification needed] and orally active derivatives were sought. A structurally similar compound, ephedrine, was identified by Japanese chemists in the Ma Huang plant and marketed by Eli Lilly as an oral treatment for asthma. Following the work of Henry Dale and George Barger at Burroughs-Wellcome, academic chemist Gordon Alles synthesized amphetamine and tested it in asthma patients in 1929. The drug proved to have only modest anti-asthma effects, but produced sensations of exhilaration and palpitations. Amphetamine was developed by Smith, Kline and French as a nasal decongestant under the trade name Benzedrine Inhaler. Amphetamine was eventually developed for the treatment of narcolepsy, post-encepheletic parkinsonism, and mood elevation in depression and other psychiatric indications. It received approval as a New and Nonofficial Remedy from the American Medical Association for these uses in 1937 and remained in common use for depression until the development of tricyclic antidepressants in the 1960s.
**question**: What was the trade name of amphetamine as a nasal decongestant?

**Answer from Dataset:**
Benzedrine Inhaler

**Response from Llama-3.1-70B:**
**reason**: The context mentions that amphetamine was developed as a nasal decongestant under a specific trade name.
**answer**: Benzedrine Inhaler

### G.3.2 BAD MIND

One case of Llama-3.1-70B's incorrect responses when $\mathcal{F}_\Omega =$ **Char-REO-INT_4**.

**Typoglycemia Prompt:**
Answer the question with word or phrase based on the context below:
**Question**: What types of tracks from popular artists did adult contemporary radio play?
**Context**: Atnoher big itpeums for the evultooin of the AC rdiao fmorat was the papultoriy of easy ltinensig or "beuafiutl msiuc" stiotnas, satitons with music saplificecly dneigesd to be purely anmebit. Wehaers most easy lniteinsg music was inttauemsnrl, caerted by rvltaieely unonwkn atstris, and rerlay pchuaesrd, AC was an aettmpt to cearte a sliimar "lite" fmorat by csooihng ceitran tcraks (both hit sneligs and album cuts) of poalupr astrits.
Response in the following format without any other information:
**reason**: reason for the answer here
**answer**: answer here

**Original Quetsion:**
**context**: Another big impetus for the evolution of the AC radio format was the popularity of easy listening or "beautiful music" stations, stations with music specifically designed to be purely ambient. Whereas most easy listening music was instrumental, created by relatively unknown artists, and rarely purchased, AC was an attempt to create a similar "lite" format by choosing certain tracks (both hit singles and album cuts) of popular artists.
**question**: What types of tracks from popular artists did adult contemporary radio play?

**Answer from Dataset:**
hit singles and album cuts

**Response from Llama-3.1-70B:**
**reason**: The context states that AC radio played certain tracks from popular artists to create a "lite" format.
**answer**: Lite tracks from popular artists

## G.4 COMMONSENSE REASONING (CSQA)

### G.4.1 GOOD MIND

One case of GPT-4o-mini's correct responses when $\mathcal{F}_\Omega =$ **Char-INS-INT_2**.

**Typoglycemia Prompt:**
Choose one choice that best answers the commonsense question below:
**Question**: WnheJre dsqoes a dqooYctor wgfork silZowtly?
**Choices**: [emergency room, nursing home, medical school, dentist, golf course]
Response in the following format without any other information:
**reason**: {reason for the choice here}
**answer**: {one choice from the choices list here}

**Original Question:**
**question**: Where does a doctor work slowly?
**choices**: [emergency room, nursing home, medical school, dentist, golf course]

**Response from GPT-4o-mini:**
**reason**: A nursing home typically involves providing ongoing care for residents, which can be a slower-paced environment compared to the fast-paced nature of an emergency room or the more specialized settings like a dentist's office.
**answer**: nursing home

**Answer from Dataset:**
nursing home

### G.4.2 BAD MIND

One case of GPT-4o-mini's incorrect responses when $\mathcal{F}_\Omega =$**Char-REO-REV**.

**Typoglycemia Prompt:**
Choose one choice that best answers the commonsense question below:
**Question**: nhoJ sehctaw eht llew desserd elpoep morf a klawtac evoba eht egats. eH snetsil ot meht kaeps desraeher senil elihw eht ecneidua snetsil. erehW si eh/ **Choices**: [theatre, new york city, fashion show, construction site, school play]
Response in the following format without any other information:
**reason**: {reason for the choice here}
**answer**: {one choice from the choices list here}

**Original Quetsion:**
**question**: John watches the well dressed people from a catwalk above the stage. He listens to them speak rehearsed lines while the audience listens. Where is he?
**choices**: [theatre, new york city, fashion show, construction site, school play]

**Response from GPT-4o-mini:**
**reason**: A "rotcod" is a doctor spelled backward, and doctors typically work in places where they provide medical care, such as an emergency room or a nursing home. However, the emergency room is specifically where doctors work in urgent situations, making it the most fitting choice. **answer**: emergency room

**Answer from Dataset:**
catwalk

## G.5 CODE GENERATION (MBPP)

### G.5.1 BAD MIND

One case of GPT-3.5-Turbo's incorrect responses when $\mathcal{F}_{\Omega} =$**Char-DEL-INT_2**.

**Typoglycemia Prompt:**
Solve the code problem below in Python:
**Problem**: W_i_e a f_n_tion to f_d e_erian n_m_er a(n, m).
Response in the following format without any other information:
**code**: {Python code here}

**Original Quetsion:**
Write a function to find eulerian number a(n, m).
**choices**: [theatre, new york city, fashion show, construction site, school play]

**Response from GPT-3.5-Turbo:**
**code**:

```python
def find_eleven_number(a, b):
    for i in range(a, b+1):
        if '11' in str(i):
            print(i)
            find_eleven
            _number(1, 100)"
```

**Answer from Dataset:**

```python
def eulerian_num(n, m):
    if (m >= n or n == 0):
        treturn 0
    if (m == 0):
        return 1
    return ((n - m) *
    eulerian_num(n - 1,
    m - 1) + (m + 1)*
    eulerian_num(n - 1, m))
```

# H  TypoP Cases

## H.1  Rectify

One case of GPT-4o's performance on *Rectify* TypoP task when $\mathcal{F}_\Omega =$**Char-REO-ALL**.

**Original Text:**
Franchising is a way for small business owners to benefit from the economies of scale of the big corporation (franchiser). McDonald's and Subway are examples of a franchise. The small business owner can leverage a strong brand name and purchasing power of the larger company while keeping their own investment affordable. However, some franchisees conclude that they suffer the "worst of both worlds" feeling they are too restricted by corporate mandates and lack true independence. It is an assumption that small business are just franchisees, but the truth is many franchisers are also small businesses, Although considered to be a successful way of doing business, literature has proved that there is a high failure rate in franchising as well, especially in UK, where research indicates that out of 1658 franchising companies operating in 1984, only 601 remained in 1998, a mere 36%.

**Typoglycemia Text:**
snaihincFrg is a way for masll eisbnssu nwoers to teiebnf mfro the oocesiemn of lcsae of the big tcprnrooaoi (hscreainfr). MdlcDaon's and ywaSbu are eepalsxm of a ceinasrfh. The llsam nsbsuies owenr can releaevg a gornts narbd name and cigsrhaupn wproe of the glearr apcomny leiwh gieenkp ireht own ievtnsmten daefoabrfl. Hvorewe, omes esaiescnfhr oelundcc thta yteh fusfer the "trwos of btoh odrwsl" eingelf tyeh are too eirrettdcs by aroprotec natamsed and lcak teru enednenicped. It is an pusostmina tath laslm seusisnb are tujs resnhfescai, but the hturt is nmya irrfsesachn are sloa mlals bessuseisn, thlhuAgo diesorendc to be a ccesfslusu way of igdon isnusbse, etluerrtai has edorpv htat rtehe is a gihh ufrilea etra in fsinnagrchi as lelw, asiyellecp in UK, eehwr ehscerra neicdsita thta out of 1658 snfrcainhig ocnpesami otienrpga in 1984, noly 601 amirened in 1998, a reem 36%.

**Rectified Text:**
Franchising is a way for small business owners to benefit from the economies of scale of the big corporation (franchiser). McDonald's and Subway are examples of a franchise. The small business owner can leverage a strong brand name and purchasing power of the larger company while keeping their own investments affordable. However, some franchisees conclude that they suffer the "worst of both worlds" feeling they are too restricted by corporate mandates and lack true independence. It is a misconception that small businesses are just franchises, but the truth is many franchisers are also small businesses. Although considered to be a successful way of doing business, literature has proved that there is a high failure rate in franchising as well, especially in the UK, where research indicates that out of 1658 franchising companies operating in 1984, only 601 remained in 1998, a mere 36%.

## H.2 SUMMARIZE

One case of Gemma-2-2B's performance on *Summarize* TypoP task when $\mathcal{F}_\Omega =$**Char-DEL-INT_1**.

**Original Text:**
A charity shop or thrift shop is a retail establishment run by a charitable organization to raise money. Charity shops are a type of social enterprise. They sell mainly used goods such as clothing, books, music albums, home videos, and furniture donated by members of the public, and are often staffed by volunteers. Because the items for sale were obtained for free, and business costs are low, the items can be sold at competitive prices. After costs are paid, all remaining income from the sales is used in accord with the organization's stated charitable purpose. Costs include purchase and/or depreciation of fixtures (clothing racks, bookshelves, counters, etc.), operating costs (maintenance, municipal service fees, electricity, heat, telephone, limited advertising) and the building lease or mortgage.

**Typoglycemia Text:**
_A c_arity s_op or thr_ft s_op is a re_ail est_blishment run by a ch_ritable organizat_on to r_ise mon_y. Char_ty s_ops are a t_pe of s_cial ente_prise. T_ey s_ll ma_nly us_d g_ods su_h as cloth_ng, bo_ks, m_sic al_ums, ho_e vid_os, and fur_iture do_ated by me_bers of the pub_ic, and are o_ten s_affed by volunte_rs. B_cause the it_ms for sa_e we_e o_tained for fr_e, and busin_ss c_sts are low, the it_ms can be s_ld at competiti_e pri_es. Aft_r cos_s are pa_d, all remai_ing in_ome f_om the s_les is u_ed in acc_rd w_th the orga_ization's s_ated charita_le pur_ose. Co_ts inc_ude pu_chase and/or deprec_ation of fix_ures (cl_thing rac_s, books_elves, counte_s, etc.), operat_ng cos_s (ma_ntenance, munic_pal se_vice f_es, ele_tricity, he_t, tele_hone, lim_ted advertisi_g) and the bu_lding lea_e or mor_gage.

**Summarized Text:**
Charity shops or thrift shops are retail establishments run by charitable organizations to raise money. They sell donated goods such as clothing, books, music albums, home videos, and furniture at competitive prices. After covering costs, including the purchase and depreciation of fixtures, operating costs, and building lease or mortgage, the remaining income is used for the organization's charitable purpose.

## H.3 TRANSLATE

One case of Gemma-2-9B's performance on *Summarize* TypoP task when $\mathcal{F}_\Omega =$**Char-REO-INT**.

**Original Text:**
A petroleum geologist is an earth scientist who works in the field of petroleum geology, which involves all aspects of oil discovery and production. Petroleum geologists are usually linked to the actual discovery of oil and the identification of possible oil deposits or leads. It can be a very labor-intensive task involving several different fields of science and elaborate equipment. Petroleum geologists look at the structural and sedimentary aspects of the stratum/strata to identify possible oil traps.

**Typoglycemia Text:**
A petrolbeum gteologist is an eLarth scientWist who woprks in the fqield of peotroleum geQology, whiSch iNnvolves all aspecrts of oil deiscovery and productcion. Puetroleum gieologists are uFsually liMnked to the acZtual discotvery of oil and the identificagtion of pUossible oil deposoits or lBeads. It can be a veCry lzabor-inteqnsive tajsk involvieng sIeveral differeHnt ficelds of scieNnce and elatborate equppiment. Peetroleum geolLogists loyok at the stRructural and sePdimentary aspeCcts of the sqtratum/strqata to idezntify possiible oil trxaps.

**Summarized Text:**
A petroleum geologist is an earth scientist who works in the field of petroleum geology, which involves all aspects of oil discovery and production. They are typically linked to the actual discovery of oil and the identification of possible oil deposits or leads. Petroleum geologists analyze the structural and sedimentary aspects of the subsurface to identify potential oil traps.

