# OpenReview forum: "MIND SCRAMBLE: UNVEILING LARGE LANGUAGE MODEL PSYCHOLOGY VIA TYPOGLYCEMIA"
_ICLR.cc/2025/Conference — Submitted to ICLR 2025_

### Official Review · Reviewer_bWR5 · 2024-10-26

**Soundness:** 1
**Presentation:** 2
**Contribution:** 2
**Rating:** 3
**Confidence:** 5

**Summary:**

This work introduces a set of Typoglycemia experiments as an example for comparing LLM behaviors to human cognitive patterns. The authors studied the scrambled text correction task (aka the known misspelling correction task) and evaluated the model on existing benchmarks while typos are there. A number of findings on how different text scrambling affects LLM performance and computational cost have been provided.

**Strengths:**

1. This work is one of the first to introduce Typoglycemia experiments (both correction and task completion) to realistic LLM benchmarks.
2. This work highlights a lot of interesting behavioral observations in LLMs dealing with misspelled input. I particularly enjoy observation 4, RQ 1, Section 4.2 (computation cost). I think a lot of the findings can give a lot of hints for future inference time reasoning research.
3. Studying LLMs using psychological methods is a timely direction to explore.

**Weaknesses:**

### Weakness 1: Overclaiming.

The most concerning problem of this work is overclaiming throughout this paper.

The authors claim that they contributed a “New Direction”.
> We propose “LLM Psychology” as an interdisciplinary framework with significant research depth, offering novel methodologies, directions and insights for the future study of LLM’s human-like cognition. To the best of our knowledge, we are the pioneer to systematically transfer cognitive psychology methodologies and experiments to LLMs, assessing the similarities and differences between LLMs and humans from a cognitive psychological perspective (Lines 112-116).

This is clearly overclaiming to me, with so much research being done in this field [1-3] ignored. I can understand that the authors try to position this work as an “example of how psychological principles can be applied to understand and evaluate LLMs”, but still, the authors also claimed that “What all of these studies lack is the systematical exploration of the generalized and intrinsic cognitive mechanisms of LLMs” (Lines 83-84). While I agree with this statement, neither did this work provide a satisfying answer for (1) a systematical/generalized study, nor “intrinsic cognitive mechanisms”, which is ill-defined and requires significant studies on mechanistic interpretability and neuroscience.

**I strongly encourage the authors to really scope down and identify their own unique angle of contributions to the community, rather than trying to convey a sensational headline to hype the community.**

---
### Weakness 2: Weak Logical Connections
There is a critical lack of logical connection between behavioral results to mechanistic cognitive principles.
Now, ignoring all the psychological stories, this paper essentially studied the scrambled text correction task (aka the known misspelling correction task) and evaluated the model on existing benchmarks while typos are there.

Some behavioral observations are interesting, though most of them are not surprising. I particularly enjoy observation 4, RQ 1, Section 4.2 (computation cost). This can give a lot of hints for future inference time reasoning research.

Some parts of the experiments can be better executed, for example, instead of “scrambling ratio”, why not just use the well-known text edit distance as an indicator?

Some parts of the experiments lack logical soundness. For example,
> Obs.3. The hidden layer representations of the same LLM across different datasets exhibit similar “cognitive patterns.” As illustrated in Figure 5, the color distributions for SQuAD and BoolQ under various FΩ appear visually similar. (Lines 522-524)

Let’s forget about the fact that the color bars in Figure 5 have different value scales for a moment. The logical connection between the “... appear visually similar” and “similar cognitive patterns” has not been discussed.

---
### Weakness 3: Missing Related Work.
The authors should discuss more on the critically related lines of work like misspelling detection, typographic attacks, typo-correction, and typo-robustness in LLMs [4-5, inter alia].

---
### Weakness 4: Communication.
Figures 1/2 certainly draw the eye with a charming simplicity reminiscent of classic, child-friendly visuals. While delightfully approachable, they might benefit from enhanced clarity and a touch more alignment with typical scientific presentation standards, as I am completely lost and need help understanding the whole pipeline and method from it. I encourage the authors to consider alternative ways of making the main figures.


[1] Hagendorff, Thilo, Thilo Hagendorff, Ishita Dasgupta, Marcel Binz, Stephanie C.Y. Chan, Andrew Lampinen, Jane X. Wang, Zeynep Akata, Eric Schulz. "Machine psychology: Investigating emergent capabilities and behavior in large language models using psychological methods." arXiv preprint arXiv:2303.13988 (2023).

[2] Demszky, Dorottya, Diyi Yang, David S. Yeager, Christopher J. Bryan, Margarett Clapper, Susannah Chandhok, Johannes C. Eichstaedt et al. "Using large language models in psychology." Nature Reviews Psychology 2, no. 11 (2023): 688-701.

[3] Li, Yuan, Yue Huang, Hongyi Wang, Xiangliang Zhang, James Zou, and Lichao Sun. "Quantifying ai psychology: A psychometrics benchmark for large language models." arXiv preprint arXiv:2406.17675 (2024).

[4] Evertz, Jonathan, Merlin Chlosta, Lea Schönherr, and Thorsten Eisenhofer. "Whispers in the Machine: Confidentiality in LLM-integrated Systems." arXiv preprint arXiv:2402.06922 (2024).

[5] Song, Gan, Zelin Wu, Golan Pundak, Angad Chandorkar, Kandarp Joshi, Xavier Velez, Diamantino Caseiro et al. "Contextual Spelling Correction with Large Language Models." In 2023 IEEE Automatic Speech Recognition and Understanding Workshop (ASRU), pp. 1-8. IEEE, 2023.

**Questions:**

Question 1: Could the authors provide more discussions on the logical connection between the observation to human cognitive patterns, as well as more psychological research citations for support in Lines 527-52 below?
> Based on these observations, we posit that the heatmap can translate each model’s unique ”cognitive pattern” through our Typoglycemia experiments, much like how different human individuals exhibit distinct cognitive patterns.

Question 2: Instead of “scrambling ratio”, why not just use the text edit distance as an indicator and visualize this pattern? Currently, the number of changes is between 1-4, what if we scale this up to large numbers of edits?

Question 3: As the authors also mentioned (line 92), LLMs’ tokenization algorithms have a lot to do with misspelling understanding and robustness under typos. In attributing these behaviors directly to the LLMs, I wondered to what extent the observations might actually stem from the tokenization process. Could the authors provide examples comparing the tokenization of correctly written versus scrambled text?

---

### Official Review · Reviewer_TBYA · 2024-10-30

**Soundness:** 2
**Presentation:** 2
**Contribution:** 2
**Rating:** 3
**Confidence:** 5

**Summary:**

This paper introduces a new approach to constructing a benchmark for LLMs based on scrambling letters words and sentences in existing benchmarks. The approach is motivated by research in psychology on reading.

**Strengths:**

The approach is creative and makes contact between psychology and the analysis of large language models.

**Weaknesses:**

1. The authors argue they are introducing “LLM psychology” as an approach despite the fact that there are a number of existing works taking this approach — the Binz and Schulz paper that is cited, their follow up work on CogBench, and more than 20 papers at the Annual Conference of the Cognitive Science Society to start with. This claim should be removed and related work should be cited more prominently.

2. The connection to psychology is very superficial. There are no direct comparisons between models and people on the same tasks or direct replications of psychological tasks.

3. As a result, it is not clear what we learn from the analyses presented in the paper. The main finding is that models are surprisingly robust to perturbation of characters, words, and sentences in existing benchmarks, but it is not clear what the consequences of this are for understanding LLMs or using them in practice.

4. The formalism introduced in Section 3.1 largely seems unnecessary. As an illustration of this, it is not used anywhere else in the paper.

**Questions:**

I did not have questions about the paper.

---

### Official Review · Reviewer_S9Qc · 2024-11-03

**Soundness:** 2
**Presentation:** 2
**Contribution:** 2
**Rating:** 3
**Confidence:** 3

**Summary:**

This paper introduces a new interdisciplinary framework called “LLM Psychology”, which applies and extends human psychology methods to investigate the Typoglycemia phenomena of large language models.

**Strengths:**

- This paper takes an interesting angle -- From the psychology and CogSic angle and comprehensively evaluate the Typoglycemia phenomenon in LLMs.
- It tests multiple open-source and closed-source models to ensure a more thorough and comprehensive evaluation.

**Weaknesses:**

- LLM Psychology is not a new direction; there have been many recent papers in this area. Therefore, I don't think you can claim to be the pioneer in systematically transferring cognitive psychology methodologies and experiments to LLMs in your paper. There are several related works (e.g., https://arxiv.org/abs/2409.11733, https://arxiv.org/abs/2402.07282, https://arxiv.org/abs/2406.17055, https://arxiv.org/abs/2407.06004) and many more, so the authors need to have a better literature review for this paper. If you could specifically highlight your contribution regarding Typoglycemia, I would consider this a novel and valuable contribution.
- The paper includes a lot of observations, which are interesting but need to be better compressed and emphasized. Currently, it feels more like a description of the table results rather than a focused analysis.
- The paper requires a more relevant and comprehensive related works section. The authors appear to have overlooked some recent papers, such as those addressing human-like mechanisms of LLMs. For example, the similarities between LLMs and human cognitive mechanisms are not cited, even though there is a substantial body of recent work on this topic (e.g.,https://arxiv.org/abs/2402.04559). Additionally, citations are missing in discussions about the model’s cognitive intuitions in reasoning tasks. Overall, the related works section suggests that the authors may not be sufficiently familiar with the latest research in this field.

**Questions:**

Same as the weakness section. If authors can address all my concerns, I'm happy to raise my score.

---

### Official Review · Reviewer_YGdX · 2024-11-03

**Soundness:** 2
**Presentation:** 3
**Contribution:** 2
**Rating:** 3
**Confidence:** 4

**Summary:**

The authors study the concept of typoglicemia for LLMs, i.e., how LLMs perform when prompted with inputs whose words’ characters (but they go beyond to sentence level) are perturbed by shuffling, deletion, insertion or addition.
The motivating idea is that humans are quite robust to these perturbations (an attack in the machine learning robustness community); such experiments are intended to show whether LLMs have similar biases to human cognition.
The authors provide a mathematical framework to explain typoglycemia, standardise the benchmark, and run experiments on several LLMs.

**Strengths:**

The paper is well written. Math, though a bit too prolonged given the simplicity of the concepts expressed (which is not bad; the opposite), makes it clear what the authors' relevant research questions are.
The authors report many interesting results on how LLMs fail when prompted with perturbed inputs and draw an interesting analogy between humans and LLMs as cognitive systems. If I understand their idea correctly, humans are their baseline to understand whether and how LLMs differ in their reasoning process when prompted with ‘scrambled’ inputs. I applaud this idea as that may attract psychologists into an area that requires interdisciplinary expertise, with the capabilities of these models that grow every day and, for some tasks, surpass that of humans.

**Weaknesses:**

The paper lacks relevant connections to related works on the adversarial robustness of NLP models. Between 2017 and 2022, a large number of influential articles were published on methods to attack NLP classifiers and then LLMs [1, 2] (at that time, BERT and GPT-2, for example). Please notice that the second article I cite is a survey that reviews tens of works.
The authors also missed some recent articles on the robustness of LLMs against text-level perturbations [3].

The authors claim they are the first to pioneer LLM Psychology, i.e., “systematically transfer cognitive psychology methodologies and experiments to LLMs.” That is not true. There are some works in this area, e.g., [4] and related works in that paper (though I state it clearly here, that article addresses a different problem).

Apart from that, my opinion is that the authors do not elaborate enough on tokenization and a model’s internal representation. At the same time, they over-focus on the analogy between the psychology of humans and machines (but I reckon that is their intent from the beginning and the general narrative of the paper). While they measure the impact of typoglycemia on the input length (e.g., Obs. 4 (4.2)), they do not conduct an in-depth analysis at the tokenization level, i.e., where the inputs are pre-processed to be fed to a model. I understand the core idea is to measure the output of an LLM as if it were a human, but with LLMs, we need to be careful first to understand whether some phenomena are the consequence of design choices and training (e.g., positional embedding and tokenization), or they “emerge” spontaneously (in that case, your approach is interesting).

Obs 2. (4.3) Should be further elaborated. While insertion may increase a model performance as the input sequence becomes longer (e.g., an LLM has more computational time as it processes more tokens (see the paper “Think dot-by-dot” for an extreme case)), the case of deletion is interesting but no rationale is given behind this performance boost.

Obs 3. (4.4) should be further elaborated. It would have been interesting to understand the reasons behind GPT-4o's increased robustness, which seems to be common in many tasks (it can be the training data as well as a more noise-resistant tokenizer/encoding).

Minor:
Figure 1 is difficult to read (it is too small; I would either simplify it a bit or enlarge it), but it is quite illustrative!


[1] Bad Characters: Imperceptible NLP Attacks, Nicholas Boucher et al.
[2] A Survey of Adversarial Defences and Robustness in NLP, Shreya Goyal et al.
[3] Are Large Language Models Really Robust to Word-Level Perturbations?, Haoyu Wang et al.
[4] Cognitive Effects in Large Language Models, Jonathan Shaki et al.

**Questions:**

1) I do not understand whether the authors are moving toward testing the (adversarial) robustness of LLMs with psychologically grounded tests. In that case, I think the idea is interesting, but I would ask them to elaborate on this.

2) How does their work connect with the large amount of literature on adversarial robustness for NLP? What is new in their approach that differs, for example, from the many works on character level perturbation of NLP classifiers?

3) How do the authors measure when a text is not intelligible anymore for humans? How do LLMs relate to that “lower bound”?

**Details Of Ethics Concerns:**

No ethical review

---

### Official Review · Reviewer_sth7 · 2024-11-04

**Soundness:** 3
**Presentation:** 2
**Contribution:** 2
**Rating:** 3
**Confidence:** 4

**Summary:**

This paper proposes to evaluate language models from the Typoglycemia perspective to verify how much alignment there is with humans in the emerging field of LLM Psychology. Specifically, the authors design experiments by formatting the original text such as changing character order (or word, and sentence order). Results across several LLMs including GPT and Gemma suggest that similar to humans, LLMs perform worse and may take longer time on tasks such as GSM8k, BoolQ, and CSQA.

**Strengths:**

1. This paper studies an interesting research question and aims to understand why LLMs are not robust to deep reasoning from a psychology perspective. This could inspire more researchers in interdisciplinary fields to explore how to evaluate and interpret language models from their expertise.
2. The paper evaluates several language models across different setups, and shows insightful observations and findings.

**Weaknesses:**

1. Although the studied research question can be inspiring, the proposed method and conclusion has been widely studied in the NLP field, More importantly, some of the findings seems to be a bit superficial. For instance, it is not convincing that showing the hidden layer representation (especially from an embedding API) would reveal that relying on data-driven mechanisms would limit the capacity for deep reasoning. This is certainly an important question for the whole LLM development, but without considering tokenization (which is a critical part in the paper's method but is not discussed), training data, and other aspects, the experiments and results would not be sufficient. Therefore, this paper may not benefit a lot of the researchers in the ICLR community.
2. Although the illustrates with a clear example, and is generally well organized, adding all these equations (from 1 - 9) to explain very simple and well adopted concepts (such as accuracy metrics) makes it unnecessarily complicated to follow.

**Questions:**

1. In Obs.2., the authors claim that LLMs' performance is affected more with tasks that require stronger reasoning capabilities. However, would manipulating, especially on the word and sentence level change the original task? How do you ensure that after TypoC and TypoP changes, the task and answers remain the same?
2. Why would prompt tokens increase after TypoFunc? The computation time should be highly relevant to the generated tokens, rather than the prompt. Furthermore, is Figure 3 showing the ratio between completion time / prompt tokens, or such ratio before and after processing the TypoFunc? The legend and the figure comments are confusing.
3. Is cosine similarity only used to compare the difference of embedding before and after the text processing step? It is very confusing to consider this as a "metric" in explanation how you run your experiments.

---

### Meta-Review · Area_Chair_o78X · 2024-12-11

**Metareview:**

The contributions of this paper were not judged as novel and well grounded in literature by initial reviews -- authors did not engage in rebuttal or discussion.

**Additional Comments On Reviewer Discussion:**

none

---

### Decision · Program_Chairs · 2025-01-22

Reject